# Vocal communication is tied to interpersonal arousal coupling in caregiver-infant dyads

**Sam Wass[1]\*, Emily Phillips[1], Celia Smith[2], Elizabeth OOB Fatimehin[1], Louise Goupil[3]**

[1]Department of Psychology, University of East London, London, United Kingdom; [2]Institute of Psychiatry, Psychology & Neuroscience, King's College London, London, United Kingdom; [3]Université Grenoble Alpes, Grenoble, France

**Abstract** It has been argued that a necessary condition for the emergence of speech in humans is the ability to vocalise irrespective of underlying affective states, but when and how this happens during development remains unclear. To examine this, we used wearable microphones and autonomic sensors to collect multimodal naturalistic datasets from 12-month-olds and their caregivers. We observed that, across the day, clusters of vocalisations occur during elevated infant and caregiver arousal. This relationship is stronger in infants than caregivers: caregivers vocalisations show greater decoupling with their own states of arousal, and their vocal production is more influenced by the infant's arousal than their own. Different types of vocalisation elicit different patterns of change across the dyad. Cries occur following reduced infant arousal stability and lead to increased child-caregiver arousal coupling, and decreased infant arousal. Speech-like vocalisations also occur at elevated arousal, but lead to longer-lasting increases in arousal, and elicit more parental verbal responses. Our results suggest that: 12-month-old infants' vocalisations are strongly contingent on their arousal state (for both cries and speech-like vocalisations), whereas adults' vocalisations are more flexibly tied to their own arousal; that cries and speech-like vocalisations alter the intra-dyadic dynamics of arousal in different ways, which may be an important factor driving speech development; and that this selection mechanism which drives vocal development is anchored in our stress physiology.

**\*For correspondence:** s.v.wass@uel.ac.uk

**Competing interest:** The authors declare that no competing interests exist.

## Editor's evaluation

This study investigates how caregiver-infant communication is situated within (and drives) fluctuations in autonomic arousal using a cutting-edge methodology that combines day-long physiological measures and audio sampling. The authors report solid evidence on how caregiver and infant vocalisation in one-year-olds cluster around moments of heightened infant (and to a lesser extent caregiver) arousal. Overall, the article highlights the importance of examining physiological arousal in the study of caregiver-infant communication and speech development. The valuable descriptive findings and the potential of the novel methods used should be of interest to readers in the field of developmental science.

## Introduction

Infants explore their vocal possibilities from birth, producing vocalisations that are on a continuum from cries (rough sounds with a high amplitude and fundamental frequency) to speech-like vocalisations, or protophones (sounds whose morphological and spectro-temporal features resemble speech

sounds) (*Kent et al., 1987*; *Nathani et al., 2006*; *Oller et al., 2019*). This pre-linguistic phase of vocal exploration is thought to be crucial for the emergence of speech: it could serve as a base for a selection mechanism whereby caregivers' differential responses to their infants' vocal outputs (e.g. different responses to cries versus protophones) progressively lead to the prioritisation of speech signals for communication, both at developmental and evolutionary scales (*Ghazanfar and Zhang, 2016*; *Locke, 2006*; *Oller and Griebel, 2020*). Temporal contingencies (*Yoo et al., 2018*) could be especially important for infants, allowing them to realise through repeated interactions that some sounds are privileged communicative signals that are particularly efficient to engage their social partners in conversation.

But what determines when infant vocalisations occur initially, and what their acoustic characteristics are? Are infants' early vocal explorations constrained, and if so, how? One possibility is that vocal explorations follow a stochastic regime early on, and that infants' explorations of their vocal tract possibilities produce a wide and unconstrained repertoire of outputs that is then narrowed down through the parental selection mechanism described above. Consistent with this idea, Oller and colleagues have proposed that a fundamental ability that supports the emergence of speech is functional flexibility (*Oller and Griebel, 2020*; *Oller et al., 2013*). An individual has functional flexibility when at least some of their vocalisations can occur alongside variable underlying affective states, and are not tied to specific communicative functions (e.g. expressing distress). This ability is necessary for the establishment of a language system: it is because we can produce specific sounds to convey different meanings that arbitrary, symbolic systems can emerge (*Oller et al., 2013*). In short, functional flexibility is a necessary condition for arbitrariness, a key feature of words that supports the emergence of conventional symbolic systems. By contrast, non-human primate vocalisations remain largely inflexible with respect to arousal even in adulthood (*Borjon et al., 2016*) (although see *Taylor et al., 2022*).

Infants would be said to have functional flexibility if specific vocalisations that they produce (e.g. protophones) were not tied to specific communicative functions, instead occuring alonside variable affective states. Consistent with this idea, by 3 months, infants can produce speech-like vocalisations in conjunction with both positive and negative facial displays, which suggests that their vocal explorations are functionally flexible in terms of valence (*Oller et al., 2013*). It remains possible, however, that their vocalisations remain tied to other affective dimensions, in particular autonomic arousal, the fast-acting neural substrate of the body's stress response mediated by the Autonomic Nervous System (ANS) (*Cacioppo et al., 2001*; *Wass, 2020*; *Pfaff, 2018*; *Porges, 2007*). Arousal and valence vary in an orthogonal fashion (*Kreibig, 2010*), so it remains possible that infants vocalisations are tied to arousal, while remaining relatively flexible with respect to valence (e.g., vocalisations produced with both positive and negative affect could be monotonically linked to arousal).

Consistent with this hypothesis, one factor that does appear to influence vocalisation likelihood early on in development is the presence of an interactive social partner (*Baumwell et al., 1997*; *Gros-Louis et al., 2006*; *Goldstein et al., 2003*): although infants also vocalise when they are alone (*Oller et al., 2019*; *Long et al., 2020*), from the first few days of life, most infants' vocalisations cluster with parental speech in time when infants are awake and actively engaged with a partner (*Caskey et al., 2011*; *Dominguez et al., 2016*). This might suggest that infants mostly vocalise when they are aroused, in the context of social interactions in particular, and thus, that their vocalisations might remain relatively inflexible – at least with respects to states of arousal, early on in development. This assumption has not been formally tested in humans, but research with marmoset monkeys has shown that vocalisation likelihood and the acoustic properties of vocalisations are both driven by rhythmic fluctuations in the autonomic nervous system (ANS) across multiple temporal scales, in infants and in adults (*Ghazanfar and Zhang, 2016*; *Zhang and Ghazanfar, 2020*; *McFarland et al., 2020*). These influences span across temporal scales, from the temporally fine-grained (spectral features of vocalisations, in the kHz range), through to context-dependent vocalisations likelihood on the scale of minutes to hours (*Zhang and Ghazanfar, 2020*).

Relatively little research has investigated whether vocal behaviours in human infants are influenced in a similar way by fluctuations in autonomic arousal. Although a number of authors have discussed the relationship between physiological arousal and vocal behaviour, particularly in the context of infant cries (*Wolff, 1967*; *Zeskind et al., 1985*; *Wilder, 1974*), no research to our knowledge (other than work from McFarland and colleagues, discussed below) have directly measured it. Studying this is important for two reasons.

First, as mentioned above, so far, research on functional flexibility has focused on valence and correspondences between visible facial affects and vocalisations only, which limits our understanding of how and when functional flexibility emerges across development (*Oller and Griebel, 2020*). Thus, here, we investigate whether each infant's and adult's vocalisation likelihood is *overall* more contingent on arousal (either their own arousal level or their partner's). We also subdivide infant vocalisations into several types: cries (which our supplementary analyses show tend to be negative affective valence) and speech-like vocalisations (which our analyses show tend to be neutral affective valence). The functional flexibility hypothesis predicts that speech-like vocalisations should be relatively independent from arousal, while cries (alarm signals) should largely co-vary with arousal (*Altenmüller et al., 2013*). Another important aspect is that functionally inflexibility of vocalisations with regard to arousal in early infancy could be inidicative of a specific learning mechanism that underlies the development of speech over time. That is, it might be that instead of a stochastic regime, what determines the acoustic features of infants vocalisations early on is precisely fluctuations in arousal, that impact on the tension of the vocal folds, and thus, on the roughness and loudness of vocalisations (*Fitch et al., 2002*). The parental selection mechanism would then operate on a repertoire of vocalisations that is not stochastic, but grounded in physiology (*Ghazanfar and Zhang, 2016*).

Second, studying how vocalisations relate to arousal changes and autonomic arousal coupling across the dyad would deepen our understanding of how caregivers identify and respond to various types of vocalisations, leading to selective reinforcement (*Locke, 2006*; *Oller and Griebel, 2020*; *Zhang and Ghazanfar, 2016*; *Goldstein and Schwade, 2008*; *Albert et al., 2018*). Many authors have described how mimicry and vocal turn-taking behaviours play roles in socio-communicative development (*Condon and Sander, 1974*; *Schneirla, 1946*; *Lester et al., 1985*; *Wilson and Wilson, 2005*; *Fogel, 2017*), and how empathy and physiological synchrony play roles in the development of self-regulation and caregiver-child affiliative bonding (*Feldman, 2007*; *Feldman, 2006*; *Fogel, 1993*; *Ham and Tronick, 2009*); but the relationship between these two areas remains relatively underexplored. McFarland and colleagues have shown that contingent vocalisations (mother to infant and infant to mother) are more common during periods of respiratory-marked synchrony (*McFarland et al., 2020*; *McFarland, 2001*). And our own previous research has shown transient increases in caregiver-child physiological synchrony following negative affect vocalisations [42][23,38]. Traditionally, the co-regulation of arousal (i.e. management of arousal across the caregiver-child dyad) has been considered important for the early development of self-regulation (*Fogel, 2017*; *Feldman, 2007*; *Kopp, 1982*; *Beebe et al., 2016*), but it has rarely been linked to the development of communicative skills. The limited previous research in this area suggests that negative affect vocalisations are more common at high arousal states, and are more likely to elicit contingent caregiver responding (*Wass et al., 2019*; *Tronick, 2007*). But, if this is the case, and cries are more likely to elicit parental responses, then how might speech-like vocalisations (i.e. non-cries) become progressively prioritised? To answer this question, it seems crucial to examine whether non-cry vocalisations also elicit changes in arousal, arousal stability and arousal coupling across the caregiver-child dyad, and whether this is related to changes in caregivers' responses to these vocalisations; but to our knowledge no previous research has examined this.

To investigate these questions, we designed new miniaturised wearable autonomic monitors (electrocardiograms and actigraphs) and miniaturised microphones and video cameras that could be worn by infant and caregivers to obtain day-long recordings in home settings. For technical reasons our microphones recorded a 5-s sample every minute (i.e. 8% of each minute) and therefore our analyses examined only large-scale arousal changes during the 20 min before and after each vocalisation (see Methods for further discussion).

We had two main research questions. First, are caregiver and infant vocalisations as inflexible with regard to arousal as those documented in non-human primates? That is, do different types of vocalisation, such as cries and speech-like sounds, show different patterns of association with arousal across the infant-caregiver dyad? Our hypothesis was that even speech-like vocalisations remain relatively tied to fluctuations in arousal during infancy, in contrast with adulthood. Second, do spontaneously occurring vocalisations during the day co-occur with specific patterns of arousal synchrony and co-regulation? Here, our hypothesis, in line with the parental selection mechanism, was that caregivers would track infants' arousal fluctuations, and that as a consequence their vocalisations and arousal would be largely tied to their infants rather than their own.

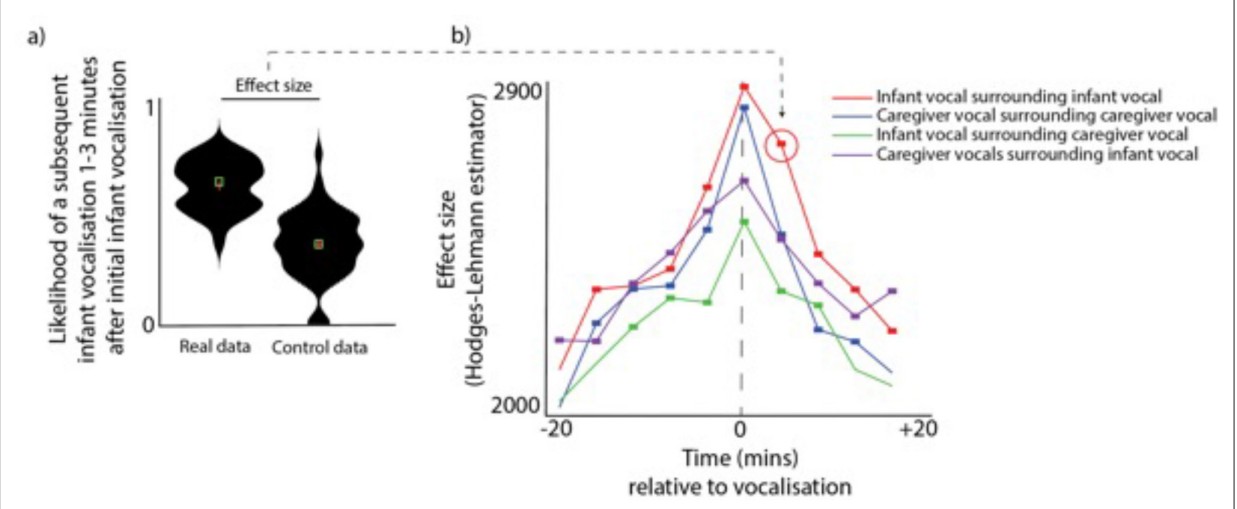

**Figure 1.** Vocalisation clusters. (**a**) Sample violin plot showing the analysis for one time interval that was then repeated iteratively across multiple time intervals in b. The plot shows the likelihood of a subsequent infant vocalisation in the time window 1–3 min following an infant vocalisation, comparing real with control data. (**b**) Same analysis repeated across multiple time windows, and across different categories. Coloured rectangles indicate time bins in which real >control after correction for multiple comparisons using a permutation-based temporal clustering procedure. Y-axis shows the Hodges-Lehman effect size of the Mann Whitney test comparing observed and control data.

## Results

Our results section is in two parts. In part 1, we analyse individual and cross-dyadic arousal changes relative to all vocalisations obtained in our data. In part 2, we subdivide vocalisations into cries and speech-like vocalisations; and, in Appendix 1, by additionally subdividing vocalisations based on vocal intensity and affective valence, manually rated by trained coders (Appendix 1 sections 2.5–2.6).

### Part 1 – All vocalisations

Our first research question was: are caregiver and infant vocalisations as inflexible with regard to arousal as those documented in non-human primates? To examine this, we conducted three analyses. First, as a preliminary analysis, we examine how vocalisations are clustered together in time. Second, we examine caregivers' and infants' arousal levels around vocalisations, using three approaches: (1) average arousal levels around vocalisations; (2) vocalisation likelihood around arousal peaks; and (3) Receiver Operator Characteristic (ROC) curves. Third, we examine arousal around vocalisations subdivided by the partner's arousal at the time of the vocalisation.

### Temporal clustering

To examine whether infants and caregivers produce clusters of vocalisations simultaneously, we performed the following analysis. For each vocalisation, we estimated the likelihood of another vocalisation occurring both before and after that vocalisation. See *Figure 1a*, which shows as an example the likelihood of a subsequent infant vocalisation occurring 1–3 min after an initial infant vocalisation. To compare the observed probabilities with chance we performed a control analysis in which we inserted random 'non-vocalisation' events into the data and repeated the analysis relative to these 'non-vocalisations', and compared the 'real' and 'control' datasets using a Mann-Whitney U test. We then repeated this analysis across multiple time windows from 20 min before the vocalisation to 20 min after. We also repeated it across multiple contrasts, looking both within an individual (e.g. infant vocalisations relative to infant vocalisations) and across a dyad (e.g. infant vocalisations relative to adult vocalisations). Multiple comparisons were corrected for using permutation-based clustering analysis (described Appendix 1 section 1.9).

After correcting for multiple comparisons, infant vocalisations were more likely to occur relative to a caregiver vocalisation across all time windows examined from 12 min prior to a vocalisation to 8 min after (all ps <0.05). This indicates that, when a caregiver vocalisation had occurred, there was an elevated likelihood of a infant vocalisation occurring for all time windows from 12 min prior to that

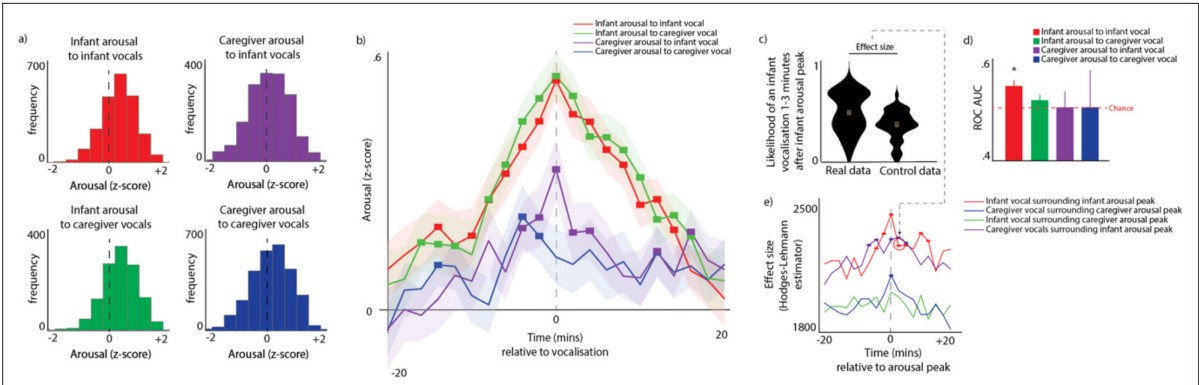

**Figure 2.** arousal changes around vocalisations. (**a**) Histogram showing the distribution of arousal levels at the time of each vocalisation. (**b**) Change in arousal levels during the period from 20 min before to 20 min after each vocalisation. Shaded areas show standard error, based on an N of 82. Coloured rectangles indicate areas in which observed >0 after correcting for multiple comparisons using a permutation-based temporal clustering procedure. (**c**) Sample violin plot showing analysis for one time interval that was then repeated iteratively across multiple time intervals in e. The plot shows the likelihood of a subsequent infant vocalisation in the time window 1–3 min following a peak in infants' arousal (10% highest values), comparing real with control data. (**d**) Receiver Operating Characteristic (ROC) Area Under the Curve (AUC) results. 0.5 shows a chance result. Error bars show the between-participant standard error of the means, based on an N of 82. * indicates significant difference from chance p<0.05, using the Mann-Whitney U test. (**e**) Same analysis as illustrated in c, repeated across multiple time windows, and across different categories. Y-axis shows the effect size of the Mann Whitney test comparing observed and control data. Coloured rectangles indicate time bins in which real >control after correction for multiple comparisons using a permutation-based temporal clustering procedure.

caregiver vocalisation to 8 min after it. Similarly, caregiver vocalisations were significantly more likely to occur relative to an infant vocalisation from 20 min before to 20 min after; infant vocalisations were significantly more likely to occur relative to another infant vocalisation from 16 min prior to a vocalisation to 20 min after (all ps <.05); and caregiver vocalisations were significantly more likely to occur relative to another caregiver vocalisation from from 16 min prior to a vocalisation to 16 min after. Overall, these findings indicate that vocalisations occurred in clusters both within an individual and across the dyad, which confirms previous reports that infant vocalisations tend to cluster with those of their social partners (; *Slone et al., 2023*).

Arousal around vocalisations. We conducted three analyses to examine the relation between arousal levels and vocalisations. First, we examined how average arousal levels change before and after vocalisations. Second, we examined how the likelihood of vocalisation changes around peak moments in arousal. Third, we calculated ROC curves to examine whether arousal levels alone can predict vocalisation likelihood.

Analysis 1 - average arousal levels around vocalisations. *Figure 2a* shows histograms of arousal levels at the time of a vocalisation, including both within-individual (e.g. infant arousal to infant vocalisations) and across the dyad (e.g. infant arousal to adult vocalisations). *Figure 2b* shows the same information, but also examines change in arousal during the period from 20 min before and 20 min after each vocalisation. *Figure 2a* is identical to the Time 0 values in *Figure 2b*. Significance testing was performed by comparing the observed arousal levels around vocalisations with a chance value of 0 (which, for z-scored data, is that individual's average arousal level across the entire day). Multiple comparisons were corrected for using a permutation-based cluster test (described Appendix 1 section 1.9). For infant arousal to infant vocalisations, significant increases in arousal were observed from 16 min before to 16 min after the vocalisation (all ps <0.05, after correction). Note that these findings are not affected by autocorrelation in the arousal data as this was removed (see Appendix 1 section 1.6). For infant arousal to caregiver vocalisations, significant increases were observed from 16 min before to 18 min after each vocalisation; for caregiver arousal to infant vocalisations, significant increases were observed from 4 min before to 6 min after; for caregiver arousal to caregiver vocalisations, no significant difference from 0 was observed at Time 0, but a significant difference was observed for the period between 4 minutes to 2 minutes before each vocalisation. Overall, these results suggest that infant arousal levels are elevated around both infant and caregiver vocalisations, and that caregivers also show smaller but significant increases in arousal around infant vocalisations.

By contrast caregivers' vocalisations show little association with their own arousal, congruent with the hypothesis of heightened vocal flexibility in human adults as compared to infants.

Analysis 2 - vocalisation likelihood around arousal peaks. Conversely, we also examined the likelihood of vocalisations occurring around peaks in arousal. We identified the moments when the infants' and the caregivers' z-scored arousal levels exceeded the top 10% most elevated values observed for that participant that day (*Figure 2c and e*). Appendix 1 section 2.4 shows the same analysis repeated with different threshold levels (5% and 20%). We then examined the likelihood of vocalisations occurring during the time windows around arousal peaks, and compared this with control data generated in the same way as described in section 1.1. Significance was calculated by performing Mann Whitney U tests and correcting for multiple comparisons using a permutation-based clustering analysis (see Appendix 1 section 1.9). Significant increases in vocalisation likelihood were observed only when we examined the likelihood of infant vocalisations around infant arousal peaks, and when we examined the likelihood of adult vocalisations around infant arousal peaks. Note that this latter finding was not significant when other threshold values were used instead (see Appendix 1 section 2.4). No significant increases in vocalisation likelihood were observed relative to adult arousal peaks. Overall, these results confirm that infant arousal peaks are associated with an increased vocalisation likelihood in both infants and (to a lesser extent) adults, but that peaks in adult arousal are not associated with increased vocalisation likelihood (a marker of greater vocal functional flexibility).

### Analysis 3

As a further test of whether arousal levels predict vocalisation likelihood differently in infants and adults, we employed a signal detection framework based on the ROC (see *Figure 2d*). Each dataset was systematically thresholded at all possible values from its minimum to maximum value. At each threshold, each epoch was individually classified either as a True Positive (above-threshold arousal, vocalisation present) or a False Positive (above-threshold arousal, vocalisation absent). If the systematic thresholding produced as many false alarms as hits, then the feature dimension could not be said to aid in predicting vocalisation likelihood. Following calculation of the ROC curves, the Area Under the Curve (AUC) was calculated: a higher AUC indicates that the feature dimension is more predictive. AUC values were calculated per participant and compared with a chance value of 0.5 using the nonparametric Mann-Whitney U test. Results indicated that the infant arousal was significantly predictive of infant vocalisation likelihood (p<0.001), but that other relationships were not. This is again consistent with the idea that infants' vocalisations are inflexibly related to their arousal.

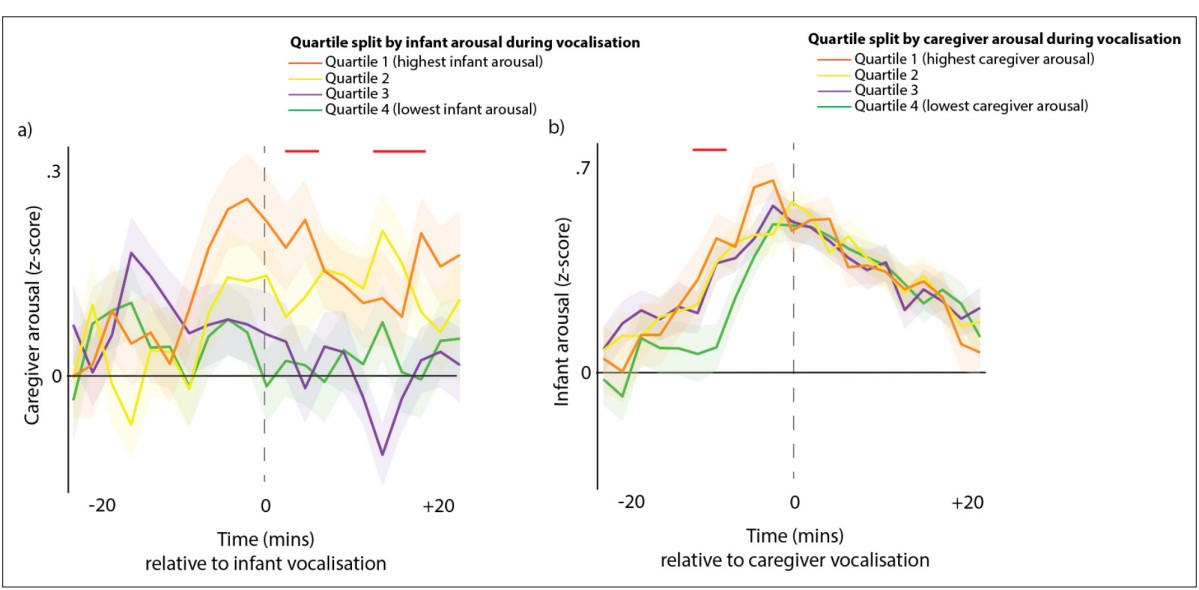

**Figure 3.** Arousal changes across the dyad following vocalisations. (**a**) Caregiver arousal subdivided by infant arousal at the time of the vocalisation. (**b**) Infant arousal subdivided by caregiver arousal at the time of the vocalisation. For all plots, shaded areas indicate standard error based on an N of 82, and red highlights indicate areas of significant difference after correction for multiple comparisons using a permutation-based temporal clustering procedure.

## Arousal around vocalisations subdivided by partner arousal at the time of vocalisation

Our final method for examining how contingent infant and caregivers' vocalisations are on arousal levels across the dyad was to subdivide all vocalisations by the partner's arousal at the time of the vocalisation. *Figure 3a* shows caregiver arousal relative to infant vocalisations (i.e. the same as the purple line from *Figure 2b*), but subdivided using a quartile split by infant arousal at the time of the vocalisation. *Figure 3b* shows infant arousal relative to caregiver vocalisation.

To estimate whether caregivers showed larger arousal changes to high arousal infant vocalisations, we performed a one-way ANOVA repeatedly for each time bin and used a permutation-based temporal clustering analysis to correct for multiple comparisons (see section 1.9). Significant effects were found ($P<0.01$) such that increased caregiver arousal was observed during the time periods 2–6 min and 10–14 min after high arousal infant vocalisations (*Figure 3a*). For infant arousal, the opposite finding was observed: high arousal caregiver vocalisations were accompanied by increased infant arousal during the period 10–6 min before the caregiver vocalisation (*Figure 3b*). Overall, these results suggest that high arousal infant vocalisations are followed by subsequent increases in caregiver arousal, and that high arousal caregiver vocalisations are preceded by increases in infant arousal.

## Control analyses

Overall, results thus far suggest that infants' vocalisations are contingent on their arousal state, whereas adults' vocalisations are independent of arousal. However, we also considered two possible alternative explanations for this finding. The first is that it may be because vocalisations are more likely to occur while the participants are in physical positions associated with increased arousal. To examine this possibility, we conducted an additional analysis in which we performed video coding to examine infants' physical position while vocalising (Appendix 1 section 2.2). In brief, this analysis suggested that 49% of infant vocalisations occurred while the infant was freely moving; 33% occurred while they were free but stationary; 7% while strapped sitting; 11% while carried. For adult vocalisations, 44% occurred while the infant was freely moving; 33% while free stationary; 10% while strapped sitting; 12% while carried. Overall when we examined how arousal levels differed by physical position we found no evidence that arousal increases around vocalisations are attributable to changes in physical position.

The second possibility is that arousal increases around vocalisations may be attributable to the physical act of vocalising itself. This may seem unlikely given that we also observed increases in infant arousal relative to caregiver vocalisations (*Figure 2b*). Yet, because we also observed that caregiver and infant vocalisations occur in clusters (*Figure 1b*), it remained possible that vocalising itself increased infant arousal in these periods. To address this, we conducted a more fine-grained analysis on a different dataset in which we continuously recorded vocalisations and arousal in 11-month-old infants and their caregivers during two 5-min tabletop interactions (see Appendix 1 section 2.3). The timings and durations of vocalisations were coded to an accuracy of 20 Hz (i.e. 50 ms), and our findings examine heart rate changes on a much finer time-scale (1 sample per second compared with 1 sample per minute for the main analyses). Overall our results suggested that, in a seated tabletop interaction, caregivers showed no change in arousal relative either to vocalisations either from themselves or their partner (the infant). Infants showed non-significant increases in arousal relative to their own vocalisations, which started to increase 5 s before a vocalisation and returned to baseline 20 s after. No changes in infant arousal were observed relative to caregiver vocalisations. The fact that arousal levels start to increase before a vocalisation suggests, consistent with animal research (*Borjon et al., 2016*), that it is unlikely that arousal changes around vocalisations are purely attributable to the physical act of vocalising itself. The fact that no changes were observed in caregiver arousal around caregiver vocalisations is also consistent with this conclusion.

## Arousal stability and arousal coupling around vocalisations

Our second research question was: do spontaneously occurring vocalisations during the day co-occur with specific patterns of arousal, arousal synchrony and arousal co-regulation? To address this we performed the calculation described in the Methods and illustrated in Figure 6.

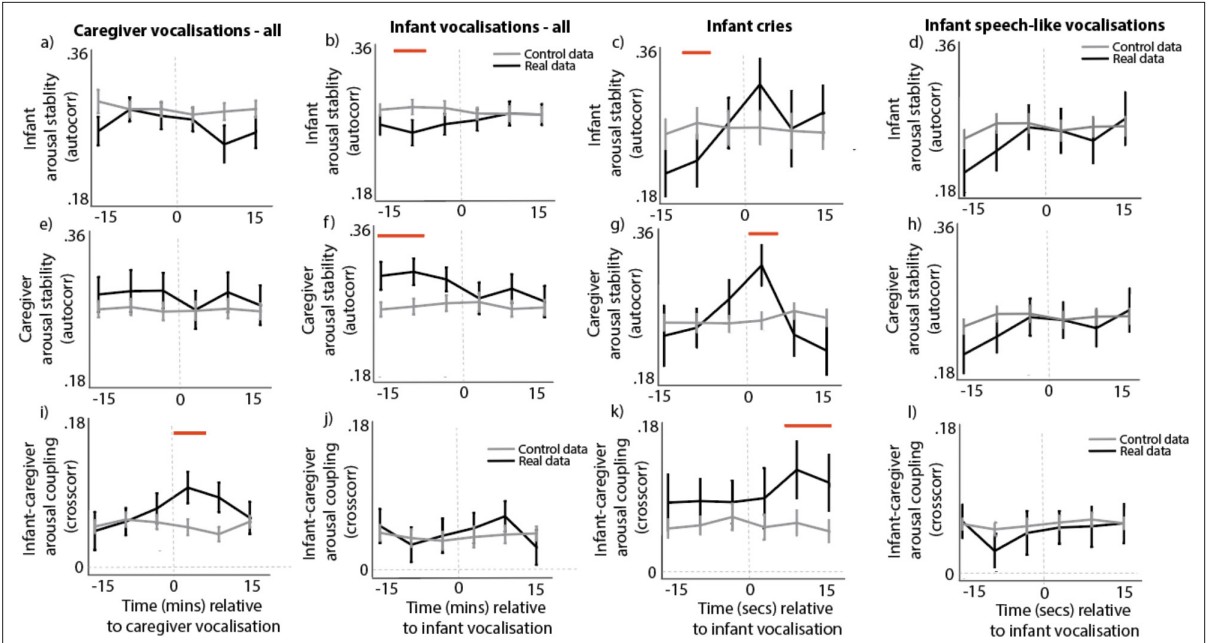

**Figure 4.** Arousal stability and coupling around vocalisations. (**a**) Infant arousal stability relative to caregiver vocalisations; (**b**) infant arousal stability relative to infant vocalisations; (**c**) infant arousal stability relative to infant cries; (**d**) infant arousal stability relative to infant speech-like vocalisations; (**e**) caregiver arousal stability relative to caregiver vocalisations; (**f**) caregiver arousal stability relative to infant vocalisations; (**g**) caregiver arousal stability relative to infant cries; (**h**) caregiver arousal stability relative to infant speech-like vocalisations; (**i**) infant-caregiver arousal coupling relative to caregiver vocalisations; (**j**) infant-caregiver arousal coupling relative to infant vocalisations; (**k**) infant-caregiver arousal coupling relative to infant cries; (**l**) infant-caregiver arousal coupling relative to infant speech-like vocalisations. Black shows the real data; grey shows the control data. Error bars show the standard errors based on an N of 82 for a-h and 74 for i-l. Sections highlighted in red indicate areas of significant difference between real and control data after correction for multiple comparisons using a permutation-based temporal clustering procedure.

## Arousal stability

Arousal stability was indexed by calculating the auto-correlation in infant and caregiver arousal. No significant changes in infant and caregiver arousal stability were observed relative to adult vocalisations (*Figure 4a and e*). By contrast, infant vocalisations were associated with decreased arousal stability in infants (*Figure 4b*), and increased arousal stability in adults (*Figure 4f*), in the time windows prior to the event. These findings differ markedly, however, when we subdivide infant vocalisations into cries and speech-like vocalisations, as shown in part 2.

## Arousal coupling

To measure arousal coupling we calculated the cross-correlation in infant-caregiver arousal, as described in the Methods and illustrated in Figure 8. Results suggested that significantly increased infant-caregiver arousal coupling was observed in the time windows following an adult vocalisation. For infant vocalisations, the same directional effect was observed but results were not significant. These findings again differ markedly when we subdivide infant vocalisations into cries and speech-like vocalisations, as shown in part 2.

## Part 2 – Infant vocalisations subdivided by vocalisation type

The findings described in part 1 indicate that 12-month-old infants' vocalisations are contingent on their arousal state, whereas adults' vocalisations are independent of arousal. However, there may be important differences between cries and speech-like vocalisations or protophones, which have been argued to already be used flexibly by infants during infancy (*Oller et al., 2013*). To further test our first research question, therefore, we examined whether different types of vocalisation, such as cries and speech-like sounds, show different patterns of association with arousal. To examine this, we recorded arousal changes relative to vocalisations subdivided by infant vocalisation type, differentiating between cries and speech-like vocalisations (see Methods).

## Temporal clustering

To examine whether infants and caregivers produce clusters of vocalisations differently as a function of vocal type, we performed the same analysis as described for part 1, this time splitting cries and speech-like vocalisations. A significant increase in the likelihood of another infant vocalisation occurring was observed from −20 min to +20 min after each infant speech-like vocalisation. For cries, a significantly increased likelihood of a subsequent vocalisation was observed for all time intervals from −20 min to +16 min. For caregiver vocalisations following infant speech-like vocalisations, significant differences from the control were observed from −4 min to +12; for caregiver vocalisations following infant cries, from −4 min to +8 min. Overall, these results suggest that in naturalistic data, vocalisations occur in clusters around both speech-like vocalisations and cries, which is inconsistent with previous reports based on laboratory recordings which suggested that infants often produce speech-like vocalisations that are not directed to social partners (*Long et al., 2020*).

## Arousal around vocalisations

To examine how arousal levels changed relative to cries and speech-like vocalisations, we performed the same three analyses as described for part 1.

### Analysis 1 - average arousal levels around vocalisations

The peak arousal (Time 0) at the time of the vocalisation was z-score.56 for cries and.41 for speech-like vocalisations, which for both categories was significantly higher than chance (both ps <0.001) (*Figure 5c*). A separate Mann-Whitney U test indicated arousal at the time of the vocalisation was significantly higher for cries than for speech-like vocalisations (p<0.01). However, arousal levels after the vocalisation regress to baseline levels more rapidly following cries than following speech-like vocalisations (*Figure 5d*). For speech-like vocalisations, significant increases in infant arousal were observed from −20 min to +20 min after (*Figure 5c*); for cries, significant increases in infant arousal were observed from −20 min to 10 min after. Significant increases in caregiver arousal were observed around infant cries (from −1 to +2 min) but not infant speech-like vocalisations.

### Analysis 2 - vocalisation likelihood around arousal peaks

Conversely, we also found that both cries and speech-like vocalisations are significantly more likely to occur during the time periods around infant arousal peaks, defined as the top 10% most elevated values observed for that participant that day (*Figure 5b*). Speech-like vocalisations were significantly more likely to occur from 3 min before to 5 min are infant arousal peaks. Cries were more likely to occur up to 3 min following an infant arousal peak.

### Analysis 3 – ROC curves

Results indicated that the infant arousal was significantly predictive of infant cries and speech-like vocalisations (both ps <0.001), but that caregiver arousal was not significantly predictive of either vocalisation type (*Figure 5e*).

Overall, these results suggest that both cries and speech-like vocalisations are associated with increases in infant arousal, but that infant arousal at the time of the vocalisation is higher for cries than speech-like vocalisations. However, speech-like vocalisations lead to more long-lasting increases in arousal. Adults show arousal changes to cries but not infant speech-like vocalisations.

## Arousal stability and arousal coupling around vocalisations

Our final analyses return to research question 2, subdividing infant vocalisations into cries and speech-like vocalisations. Our aim was to answer the question: do spontaneously occurring cries and speech-like vocalisations during the day co-occur with specific patterns of arousal, arousal synchrony and arousal co-regulation?

### Arousal stability

Infant cries were accompanied by reduced infant arousal stability in infants in the time window prior to the event (*Figure 4c*), and increased caregiver arousal stability in the time window following the event (*Figure 4g*). No changes were observed around infant speech-like vocalisations.

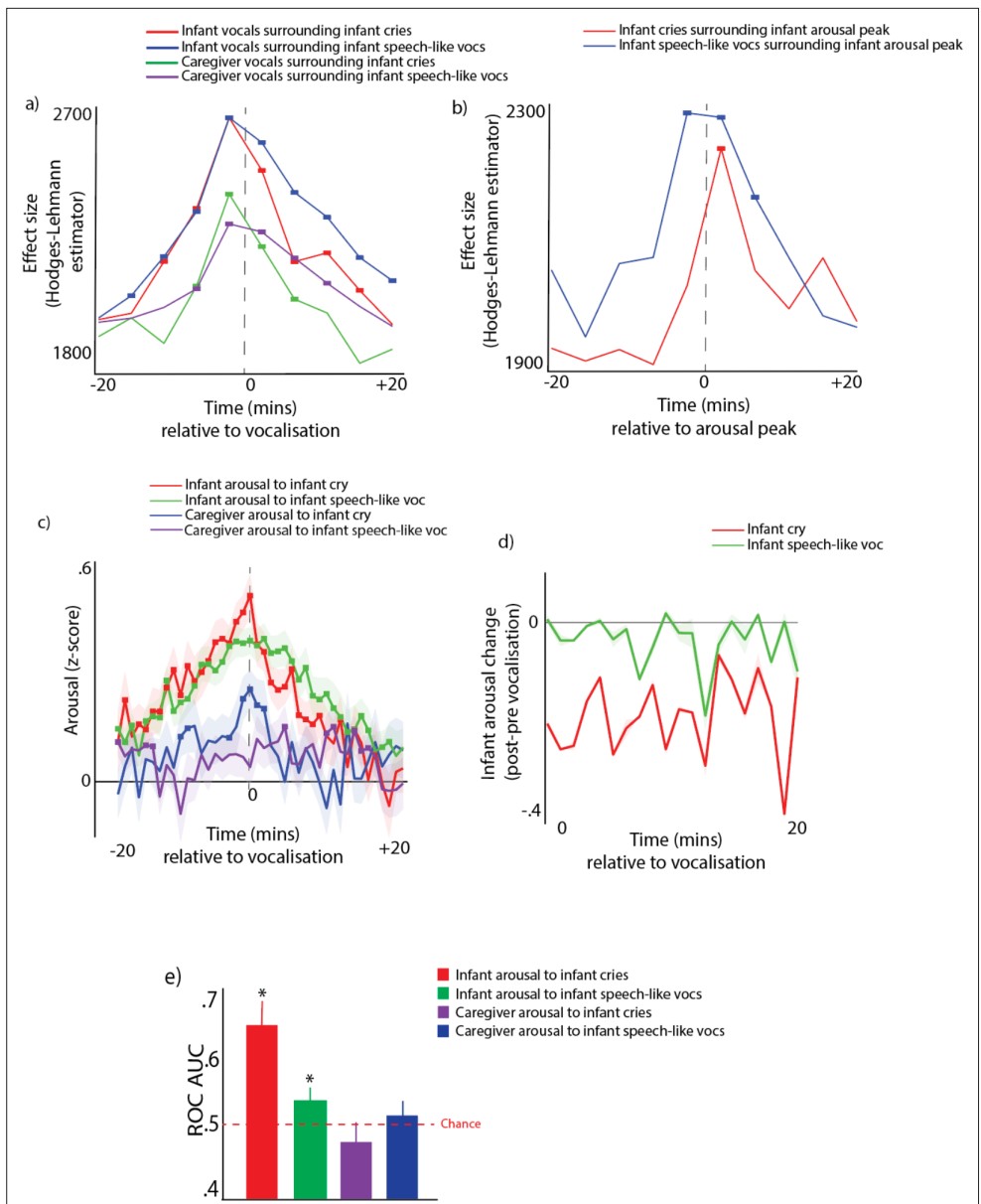

**Figure 5.** Vocalisation clusters and arousal around vocalisations subdivided by infant vocalisation type.
(**a**) Likelihood of infant and caregiver vocalisations during the time period before and after known infant vocalisations. (**b**) Likelihood of infant cries and speech-like vocalisations during the time period relative to infant 90th centile arousal peaks. (**c**) Change in arousal levels relative to vocalisations. Shaded areas show the standard errors based on an N of 82. For all plots, coloured rectangles indicate time windows in which real >control after correction for multiple comparisons using a permutation-based temporal clustering procedure. (**d**) Plot showing same data as 5 c, but showing pre- vs post-vocalisation differences in arousal around cries and speech-like vocalisations. Values above 0 indicate that post vocalisation arousal >pre vocalisation arousal. (**e**) Receiver Operating Characteristic (ROC) Area Under the Curve (AUC) results. 0.5 shows a chance result. Error bars show the between-participant standard error of the means based on an N of 82. * indicates significant difference from chance p<0.05, * indicates significant difference from chance p<0.05, using the Mann-Whitney U test.

## Arousal coupling

For cries, increased caregiver-infant arousal cross-correlations were observed in the time windows following a cry (*Figure 4k*). No changes were observed for speech-like vocalisations.

Overall, these results suggest that infant cries are accompanied by decreased infant arousal stability prior to the vocalisation and increased caregiver-child arousal coupling after the vocalisation;

no effects are observed for infant speech-like vocalisations. These results suggest that there are differentiations between cries and speech-like vocalisations with respects to arousal co-regulation across the dyad.

### Additional analysis: infants' vocal affects and caregivers' vocal types

In Appendix 1 we present additional analyses to further investigate our second research question, which is to examine how different types of vocalisation play different roles in arousal co-regulation across the infant-caregiver dyad. To do this, we subdivided infant and caregiver vocalisations by vocalisation affect, intensity and type, on the basis of a manual rating of the data by trained coders (Appendix 1 sections 2.5, 2.6). Note that the vocalisations identified as negative mostly corresponded to cries (98%), and positive vocalisations mostly included protophones, with a mixture of quasi-resonant vowel (20%), fully-resonant vowel (21%), marginal syllable (9.6%), and canonical syllable (33%). First, we examined infant vocal affects as a function of valence**Fig S8a, S8c** and intensity (see Appendix 1 section Supplementary analyses for part 2 – arousal by infant vocalisation affect and intensity). Our results suggest that larger infant arousal changes are observed relative to negative affect, and high-intensity vocalisations, which as we report above mostly correspond to cries.

Next, we examined caregiver vocalisation intensity**Fig S9a, S9c** (see Appendix 1 section Supplementary analyses for part 2 – arousal by caregiver vocalisation type and intensity). Our results suggested that there was no relationship between caregiver arousal and caregiver vocal intensity, but that caregiver vocalisation intensity is influenced by infant arousal during the time window prior to the vocalisation.

Second, we examined caregiver vocalisation type (differentiating positive, stimulating, intrusive, and sensitive vocalisations)Fig S9b, S9d. Our results suggested that there was no relationship between caregiver arousal and caregiver vocalisation type, but that caregiver vocalisation type is influenced by infant arousal during the time window prior to the vocalisation.

## Discussion

Using day-long home recordings obtained using miniaturised wearable autonomic monitors and microphones, we examined how autonomic arousal cofluctuates with vocal behaviours in caregiver-infant dyads recorded in a naturalistic home setting. We examined within-individual relationships (e.g. how infant arousal relates to infant vocalisation likelihood) and cross-dyad relationships (e.g. caregiver arousal to infant vocalisation likelihood). We also examined how caregiver-child arousal coupling cofluctuates with vocalisation likelihood. From our results the following conclusions can be drawn:

First, the within-individual relationship between arousal and vocalisation likelihood is strong in infants, and weaker in adults (*Figure 2a and b*). Symmetrically, for infants, there is an increase in the likelihood of vocalising around arousal peaks, which is not the case for adults (see *Figure 2*). The same result was confirmed by our ROC analyses, suggesting that vocalisation timings can be predicted based on arousal levels (*Figure 2*). Our findings are based on analyses conducted at the minute-level temporal scale. Findings from a more temporally fine-grained supplementary analysis (see Appendix 1 section 2.2) suggest, consistent with findings from animal research, that it is unlikely that these changes are purely attributable to the physical act of vocalising itself. These findings are also unlikely to be attributable to changes in physical positioning around vocalisations, as our supplementary analyses suggest that most vocalisations occur while free roaming, and that differences in arousal contingent on physical positioning are limited (see Appendix 1 section Supplementary Analysis 1 – video analyses of physical position while vocalising). This suggests that infants vocalisations are still relatively inflexible with respects to states of arousal at 12 months.

Our finding that the association between arousal and vocalisation likelihood is strong in infants, but weaker in adults, can be contextualised by previous research suggesting that the association between arousal and vocalisation likelihood is present in both infant *and* adult marmoset monkeys (*Ghazanfar and Zhang, 2016*; *Borjon et al., 2016*). Whereas previous research has suggested that human infants already show vocal flexibility with respect to affective valence (i.e. they can use similar vocalisations in conjunction with various facial affects) (*Oller and Griebel, 2020*; *Oller et al., 2013*), our findings suggest that infant vocalisations are relatively *inflexible* with regard to arousal during early development. This was true both for cries and for speech-like vocalisations, which were also

more likely to occur around arousal peaks (*Figure 5b*), as confirmed by our ROC analyses (*Figure 5e*). This discrepancy with previous findings might be due to a genuine difference in functional flexibility across arousal and valence, which is possible given the orthogonality of these two constructs (one can be happy, highly aroused and positive, and angry, highly aroused and negative). However, at this stage it remains equally possible that the discrepancy between our study and previous studies focusing on facial affects stem from methodological differences. Physiological measures might be more sensitive than relying on overt displays, and in future we could use other measures (e.g. acoustic analyses of vocalisations *Fitch et al., 2002*) to try to understand whether and how functional flexibility develops at the same rate for arousal and valence. Nonetheless, our findings suggest that, in early infancy, vocal production is directly related to the arousal state of the infant, with increases in arousal necessary for infants to vocalise, whatever vocalisation type.

By contrast, our results show that caregiver vocalisation likelihood is more influenced by the infant's arousal than by the caregiver's own arousal. The likelihood of an adult vocalising increases around infant arousal peaks (*Figure 2e* – purple line), more than around the caregiver's own arousal peaks (*Figure 2e* – blue line). No relationship was observed between caregiver vocal intensity and caregiver arousal (Fig S9c), but high-intensity caregiver vocalisations are preceded by high infant arousal (Fig S9a). Similarly, no associations were noted between caregiver vocalisation type and caregiver arousal (Fig S9d), whereas positive caregiver vocalisations are more likely to be preceded by high infant arousal (Fig S9b). Finally, caregivers are more likely to produce high arousal vocalisations following an increase in infant arousal (*Figure 3b*).

Taken together, these findings confirm that caregivers' speech is flexible with respect to their own levels of arousal (in contrast with what has been documented in other primate species *Zhang and Ghazanfar, 2016*) but reveal that it is finely attuned to their child's levels of arousal. This suggests that speech is an under-appreciated mechanism for arousal co-regulation during early life. Previous research has pointed to caregiver-child arousal synchrony as a mechanism for arousal co-regulation (*Feldman, 2007*; *Tronick, 2007*). One mechanism for this might be that parents dynamically modulate their own arousal level to match their child (*Wass et al., 2019*). The present findings reinforce this by suggesting, for example, that high arousal infant vocalisations tend to be followed by subsequent increases in caregiver arousal (*Figure 3a*). Vocalisations thus appear as an important medium through which arousal is made manifest between caregivers and their infants, supporting coregulation at the level of the dyad.

Our vocal type findings also point to a clear role for vocalisations in arousal co-regulation (*Wolff, 1967*; *Zeskind et al., 1985*). Infant arousal is higher when they produce cries (*Figure 5c*), and they show decreased arousal stability around cries (*Figure 4c*). Infant cries also lead to changes across the dyad: increased caregiver arousal stability is observed in the time following the vocalisation (*Figure 4g*), as well as increased caregiver-child arousal coupling (*Figure 4k*). Comparing arousal levels before and after a vocalisation suggests that cries tend to be followed by decreases in arousal (*Figure 5c*) and increases in arousal stability (*Figure 4c*). This may be because cries are less likely to occur in clusters (i.e. to be followed by another vocalisation) than speech-like vocalisations are (*Figure 5a*). This suggests that cries are events that serve to down-regulate the infant's arousal, and that the likely mechanism for this is through caregiver-infant arousal coupling.

Speech-like vocalisations show a strikingly different profile. Although arousal levels at the time of the vocalisation are also elevated relative to baseline (*Figure 5c*), the profile of change after the vocalisation is markedly different as compared to cries. Speech-like vocalisations tend to lead to sustained increases in infant arousal, in contrast to cries which lead to decreases in arousal (*Figure 5c and b*). Infant speech-like vocalisations are also more likely to be followed by other speech-like vocalisations from both the infant and the caregiver (*Figure 5a*). This potentially indicates increased attentional processes in the time-period after a speech-like vocalisation that could support the processing of caregiver's verbal responses, information encoding, and influence later vocal production (*Zhang and Ghazanfar, 2020*; *Goldstein and Schwade, 2008*).

Most important to selective reinforcement accounts of early speech development, whereas cries vocalisations lead to changes in arousal stability and coupling across the dyad, speech-like vocalisations do not (*Figure 4h and l*). Thus, despite the fact that they are both associated with fluctuations in the infant's own arousal, there are functional differences between cries and speech-like vocalisations at the level of the dyad: compared to cries, speech-like vocalisations do not seem to relate to arousal

co-regulation, and they induce less changes in arousal in caregivers. This may be an important mechanism for selective learning in early speech development, with caregivers' unimpaired arousal allowing for more flexible, timelier, and, potentially, more semantically attuned responses to these vocalisations. Supporting this suggestion, parents overlap more frequently in their responses to infant cry-sounds, compared to protophones (*Yoo et al., 2018*), and, in our data, speech-like vocalisations were more likely to be followed by caregiver vocalisations over a longer time-period, compared to cries (*Figure 5a*). These data are consistent with the idea that a parental selection mechanisms grounded in stress physiology is the 'engine for vocal development' (*Ghazanfar and Zhang, 2016*).

Although technical factors meant that we were confined to random sampling during the day rather than continuous recordings, our analyses suggest that this sparse sampling preserves the temporal structure of the data (see Appendix 2 section Supplementary analysis – simulation to examine the effects of sparse sampling on the data); furthermore, we examine event-related changes relative to vocalisations, and so the presence of undetected vocalisations can only have weakened the patterns of event-related change that we have documented here. Nevertheless, future research based on continuous recordings would allow us to examine in more detail the role of turn-taking behaviours in communicative exchanges – examining, for example, whether arousal facilitates effective communication between caregiver and child by making children more likely to respond to verbal initiations by the caregiver. Furthermore, it would also be interesting to examine whether the link between arousal and vocalisations remains unchanged even in the absence of the caregiver, or where the caregiver is unresponsive.

In future, it would also be interesting to explore the role of vocalisations in developmental psychopathology. For example, our present results suggested that, in typical dyads, high arousal infant vocalisations tend to be followed by increases in caregiver arousal. But other work from our group has shown that, in caregivers with elevated anxiety, moments of high infant arousal were *more* likely to be accompanied by high caregiver arousal; that anxious caregivers were more likely to vocalise intensely at high arousal, and to produce intense vocalisations that occurred in clusters; and that high intensity vocalisations were associated with more sustained increases in autonomic arousal for both anxious caregivers and their infants (*Smith et al., 2021a*; *Smith et al., 2021b*). Understanding how arousal relates to vocalisation likelihood across typical and atypical development is an important goal for future research. Relatedly, longitudinal studies are needed to examine the association between infant arousal and vocalisations across development; when, for example, do speech-like vocalisations become functionally flexible from arousal, and how is this affected by early infant-caregiver coregulation?

Overall, our data show that there is a functional dissociation between speech-like vocalisations and cries: cries are more likely to lead to changes in the caregiver's arousal, while speech-like vocalisations are more likely to associate with sustained increases in infant arousal, as well as an increase in vocalisations in both infants and caregivers. These results are consistent with the idea that caregivers' differential responses to specific types of vocalisations (i.e. speech-like vocalisations), which are not yet produced flexibly by infants, may be an important factor driving speech development, and they suggest that this bidirectional physiological process supports progressively specialised vocalisations through parental selection.

## Materials and methods
### Experimental participant details

The project was approved by the Research Ethics Committee at the University of East London (Approval number: EXP 1617 04). Informed consent, and intent to publish, were obtained in the usual manner. Participants were recruited from the London, Essex, Hertfordshire and Cambridge regions of the UK. In total, 91 infant-caregiver dyads were recruited to participate in the study, of whom usable autonomic data were recorded from 82. Of these, usable paired autonomic data (from both caregiver and child) were obtained from 74 participants. Further details, including exclusion criteria, and detailed demographic details on the sample, are given in Appendix 1 section 1.1. The sample size was selected following power calculations presented in the original funding application ES/N017560/1. Of note, we excluded families in which the primary day-time care was performed by the male caregiver, because the numbers were insufficient to provide an adequately gender-matched sample. All participating

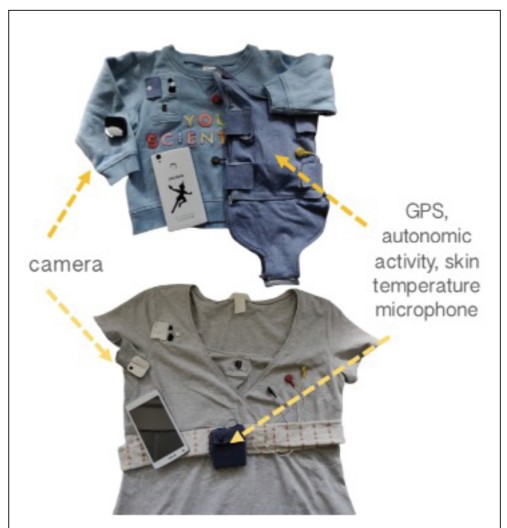

**Figure 6.** Photographs of recording equipment used.

caregivers were, therefore, female. Participants received £30 in gift vouchers as a token of gratitude for participation, split over two visits.

## Experimental method details

Participating caregivers were invited to select a day during which they would be spending the entire day with their child but which was otherwise, as far as possible, typical for them and their child. The researcher visited the participants' homes in the morning (c. 7.30 - 10am) to fit the equipment, and returned later (c. 4 - 7pm) to pick it up. The mean (std) recording time per day was 7.3 (1.4) hr.

The equipment consisted of two wearable layers, for both infant and caregiver (see *Figure 6* and *Figure 7*). For the infant, a specially designed baby-grow was worn next to the skin, which contained a built-in Electrocardiogram (ECG) recording device (recording at 250 Hz), accelerometer (30 Hz), Global Positioning System (GPS) (1 Hz), and microphone (11.6 kHz). A T-shirt, worn on top of the device, contained a pocket to hold the microphone and a miniature video camera (a commercially available Narrative Clip 2 camera). For

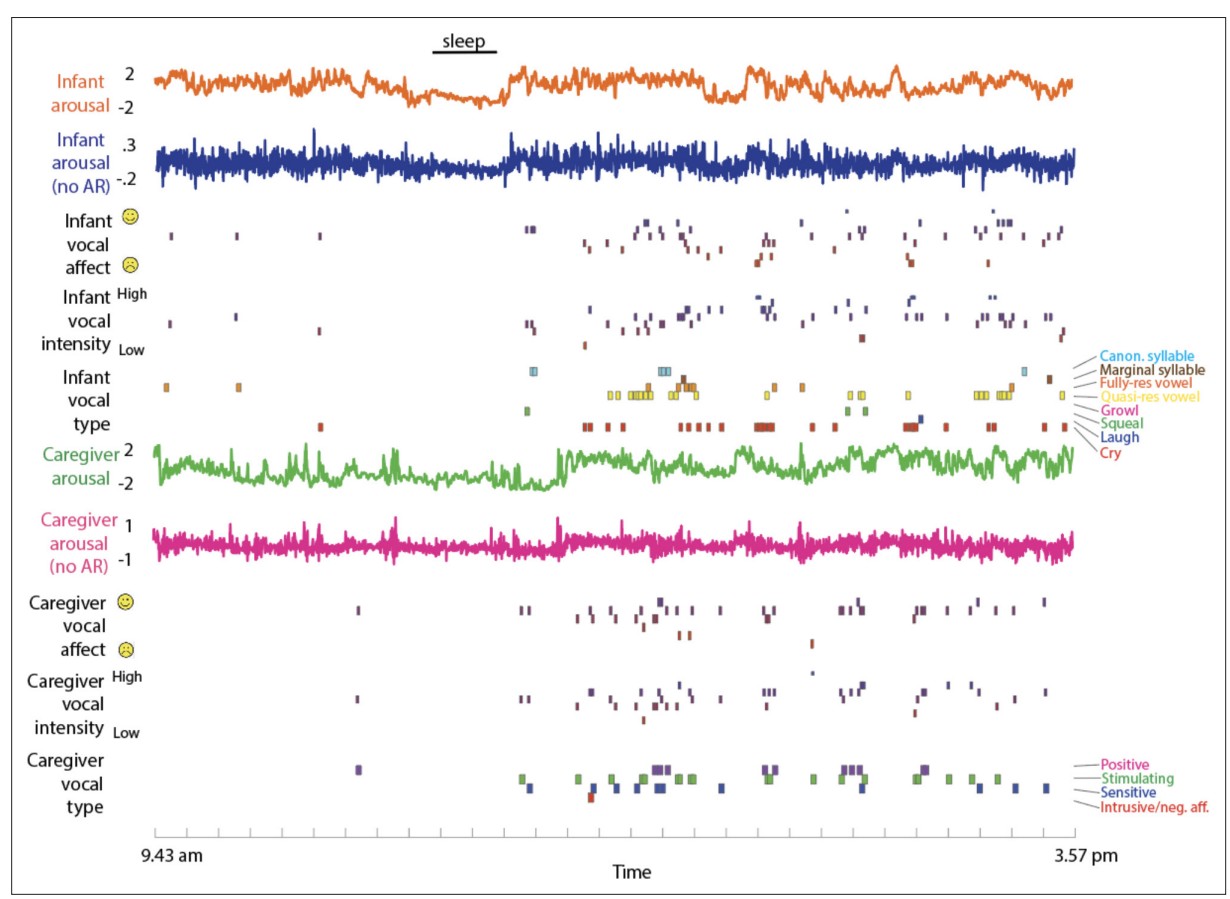

**Figure 7.** Raw Data Sample. from top to bottom: infant arousal composite score (see SM sections 1.2–1.5); infant arousal after removal of the auto-correlation (see SM section 1.6); infant vocal affect (see Methods section); infant vocal intensity; infant vocalisation type; caregiver arousal; caregiver arousal after removal of the auto-correlation; caregiver vocal affect; caregiver vocal intensity; caregiver vocalisation type.

the caregiver, a specially designed chest strap was also worn next to the skin, containing the same equipment. A cardigan, worn as a top layer, contained the microphone and video camera. The clothes were comfortable when worn and, other than a request to keep the equipment dry, participants were encouraged to behave exactly as they would do on a normal day.

At the start and end of each recording session, before the devices were inserted into the clothes worn by the participants, the researchers synchronised the two devices by holding them on top of one another and moving them sharply from side to side, once per second for 10 consecutive seconds. Post hoc trained coders identified the timings of these movements in the accelerometer data from each device independently. This information was used to synchronise the two recording devices.

## Quantification and statistical analysis

### Autonomic data parsing and calculation of the autonomic composite measure

Further details on the parsing of the heart rate (Appendix 1 section 1.2), heart rate variability (Appendix 1 section 1.3), and actigraphy (Appendix 1 section 1.4) are given in Appendix 1. In Appendix 1 section 1.5 we present our motivation for collapsing these three measures into a single composite measure of autonomic arousal (Figure S3). In section 1.6 we present a description of how the autocorrelation was removed from the arousal data.

### Home/Awake coding

Our preliminary analyses suggested that infants tended to be strapped-in to either a buggy or car seat for much of the time that they were outdoors, which strongly influenced their autonomic data. For this reason, all of the analyses presented in the paper only include data segments in which the dyad was at home and the infant was awake. A description of how these segments were identified are given in Appendix 1 (section 1.7). Following these exclusions, the mean (*std*) total amount of data available per dyad was 3.7 (*1.7*) hr, corresponding to 221.5 (*102.4*) 60 second epochs per dyad.

### Vocal coding

The microphone recorded a 5 s snapshot of the auditory environment every 60 s. *Post hoc*, trained coders identified samples in which the infant or caregiver was vocalising, and the following codings were applied. For each coding scheme, consistency of rating between coders was achieved through discussions and joint coding sessions based on an *ersatz* dataset, before the actual dataset were coded. All coders were blind to study design and hypothesised study outcome.

Importantly, analyses conducted on a separate, continuous dataset (see Appendix 1, section S10) suggest that the temporal structure of our vocalisations was maintained despite this 'sparse sampling' approach. Furthermore, our analyses examine how arousal changes relative to observed vocalisations, and any arousal changes that we do observe time-locked to vocalisations would be weakened (not strengthened) by the fact that the vocalisation data were sparsely sampled (because power would have been reduced by missing vocalisations through the sparse sampling method, rather than increased).

Infant data. (*i*) *vocalisation type*. A morphological coding scheme (*Oller et al., 2013*) was applied with the following categories: cry, laugh, squeal, growl, quasi-resonant vowel, fully-resonant vowel, marginal syllable, canonical syllable. Overall, 29% of vocalisations were cries; 1% laughs; 1% squeal; 3% growl; 18% quasi-resonant vowel; 18% fully-resonant vowel; 6% marginal syllable; 23% canonical syllable. For analyses presented in the main text these were collapsed into cries and speech-like vocalisations, which included the following non-cry categories: quasi-resonant vowel; fully-resonant vowel; marginal syllable; canonical syllable. Laughs, squeals and growls were excluded due to rarity. (ii) *vocal affect* was coded on a three-point scale for vocal affect (negative (fussy and difficult)), neutral or positive (happy and engaged). In order to assess inter-rater reliability, 11% of the sample was double coded; Cohen's kappa was 0.70, which is considered substantial agreement (*McHugh, 2012*). (iii) *vocal intensity* was coded on a three-point scale from low emotional intensity, neutral, or high emotional intensity.

Adult data. (i) *vocalisation type*. A trained coder listened to vocalisations one by one and categorised them into the following categories: Imperative, Question, Praise, Singing, Imitation of Baby Vocalisation, Laughter, Reassurance, Sighing, Storytelling. These were then further collapsed into four supraordinate categories: Positive (Singing, Laughter); Stimulating (Question); Intrusive/negative

affect (Imperative, Sighing); Sensitive (Praise, Imitation of Baby Vocalisation, Reassurance, Story-telling). Overall, 14% of vocalisations were Positive; 30% were Stimulating; 41% were Intrusive; 15% were Praise. In addition, (ii) *vocal affect and* (iii) *vocal intensity* were coded in the same way as for the infant data. In order to assess inter-rater reliability, 24% of the sample was double coded; Cohen's kappa was 0.60, which is considered acceptable (*McHugh, 2012*).

### Physical positions while vocalising
We also ascertained the physical position of our participants while vocalising (Appendix 1 section 1.8).

### Permutation-based temporal clustering analyses
To estimate the significance of time-series relationships, a permutation-based temporal clustering approach was used. This procedure, which is adapted from neuroimaging (*Maris and Oostenveld, 2007*; *Maris, 2012*), allows us to estimate the probability of temporally contiguous relationships being observed in our results, a fact that standard approaches to correcting for multiple comparisons fail to account for (*Maris, 2012*) (see also *Oakes et al., 2013*). See further details in Appendix 1 section 1.9.

### ROC analyses
In order to assess the selection of visual features we employed a signal detection framework based on the Receiver Operator Characteristic (ROC). This analyses the degree to which arousal levels predict the timings of vocalisations relative to the timings of randomly sampled comparison samples, epoch by epoch. See Results section and (*Fawcett, 2006*) for more details.

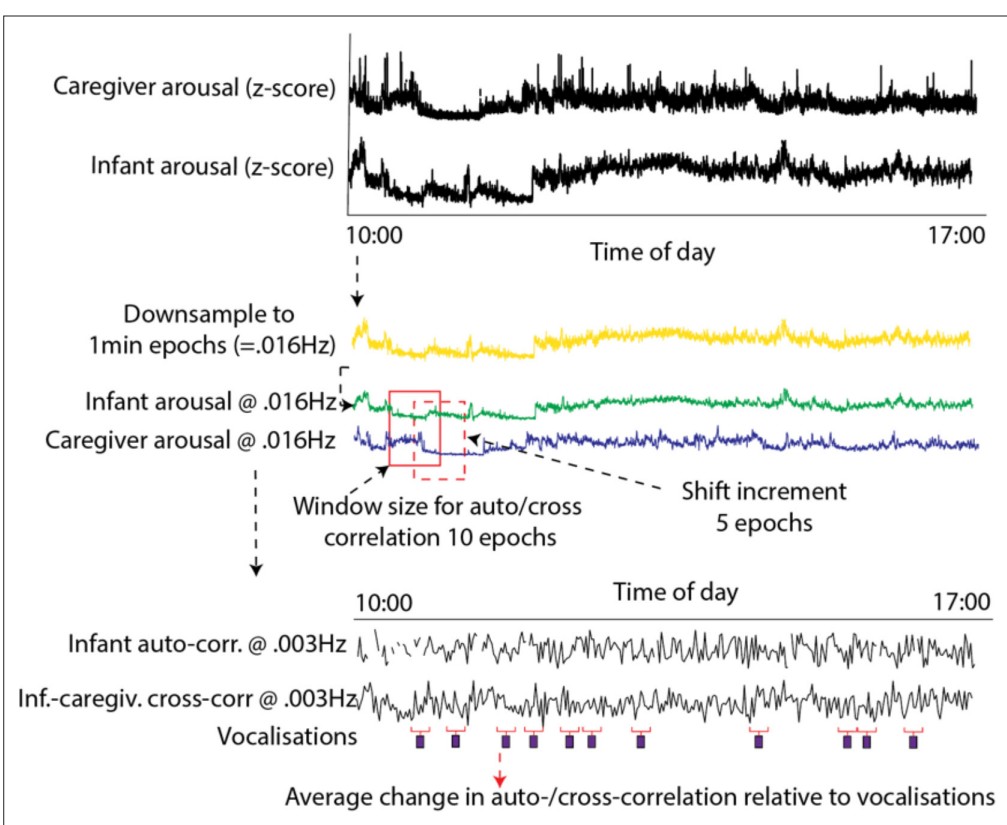

**Figure 8.** Schematic illustrating the auto- and cross-correlation analyses. Arousal data were downsampled to 1 min epochs (corresponding to the sampling frequency of the microphone data). The windowed auto- and cross-correlation was then calculated, using a window size of 10 epochs, which shifted 5 epochs between windows. The average change in auto- and cross-correlation relative to vocalisations was then calculated.

## Arousal stability

Arousal stability was measured by calculating the auto-correlation in infant and caregiver arousal, considered separately. The auto-correlation was calculated using the Matlab function nanautocorr.m, written by Fabio Oriani. Only the first lag term was reported as previous analyses have shown that autocorrelation data show a strong first order autoregressive tendency (*Wass et al., 2016*).

## Arousal coupling

Arousal coupling was measured by calculating the zero-lag cross-correlation between infant and caregiver arousal. The cross-correlation was calculated by first applying a linear detrend to each measure independently and then calculating the Spearman's correlation between the infant and caregiver arousal data within that window.

## Moving window analyses

To estimate how stability and coupling changed relative to vocalisations, we used a moving window analysis (see *Figure 8*). Arousal data were downsampled to 1-min epochs (0.016 Hz) (which was the sampling frequency of our microphone data). The size of the moving window was set arbitrarily at 10 epochs, with a shift of 5 epochs between windows. We excerpted the stability and coupling values around each individual vocalisation, and averaged these across all vocalisations.

## Control analysis

Participant by participant, for each vocalisation that was observed, a random 'non-vocalisation' moment was selected as a moment during the day when the dyad was at home and the infant was awake but no vocalisation occurred. The same moving window analysis described above was then repeated to examine change relative to this 'non-vocalisation event'. The same procedure was repeated 1000 times and the results averaged. Real and observed data were compared using the permutation-based temporal clustering analyses described above.

# Acknowledgements

This research was funded by ESRC grant number ES/N017560/1, by ERC grant number ONACSA 853251, by Project Grant RPG-2018–281 from the Leverhulme Trust and by an ERC Marie Curie Fellowship JDIL 845859. Thanks to Kaili Clackson and Farhan Mirza for help with data collection; to Caitlin Gibbs, Emma Bruce-Gardyne, Florian Andrey-Csolm, Joan Eitzenberger, Leanne Barnes, Louise Stubbs, Deborah Scnatlebury and Anne Hepworth for help with data coding. Thanks to members of the UEL BabyDev Lab for comments and discussions on earlier drafts of this manuscript, and to all participating children and caregivers.

## Additional information

### Funding

| Funder | Grant reference number | Author |
|---|---|---|
| Economic and Social Research Council | ES/N017560/1 | Sam Wass |
| European Research Council | JDIL 845859 | Louise Goupil |
| European Research Council | ONACSA 853251 | Sam Wass |
| Leverhulme Trust | RPG-2018–281 | Sam Wass |

The funders had no role in study design, data collection and interpretation, or the decision to submit the work for publication.

## Author contributions
Sam Wass, Conceptualization, Data curation, Formal analysis, Funding acquisition, Investigation, Visualization, Methodology, Writing – original draft, Project administration, Writing – review and editing; Emily Phillips, Formal analysis, Writing – original draft, Writing – review and editing; Celia Smith, Investigation, Methodology; Elizabeth OOB Fatimehin, Formal analysis; Louise Goupil, Writing – original draft, Writing – review and editing

## Author ORCIDs
Sam Wass ⓘ http://orcid.org/0000-0002-7421-3493
Elizabeth OOB Fatimehin ⓘ http://orcid.org/0000-0002-5179-9583

## Ethics
Human subjects: The project was approved by the Research Ethics Committee at the University of East London (Approval number: EXP 1617 04). Informed consent, and intent to publish, were obtained in the usual manner.

## Decision letter and Author response
Decision letter https://doi.org/10.7554/eLife.77399.sa1
Author response https://doi.org/10.7554/eLife.77399.sa2

---

# Additional files

## Supplementary files
• Transparent reporting form

## Data availability
Due to the personally identifiable nature of this data (home voice recordings from infants) the raw data is not publicly accessible. Researchers who wish to access the raw data should email the lead author s.v.wass@uel.ac.uk. Permission to access the raw data will be granted as long as the applicant can guarantee that certain privacy guidelines (e.g. storing the data only on secure, encrypted servers, and a guarantee not to share it with anyone else) can be provided. In order to allow access to the raw data the name of the applicant will also need to be added to our current ethics approval from the University of East London. This is expected to be routine, as long as the applicant is able to provide these guarantees. De-identified versions of the data - i.e. the processed Autonomic Nervous System data, and the raw coding showing the timings of when vocalisations were recorded, is available here: https://doi.org/10.5061/dryad.612jm6473. The code used to conduct the analyses is available here: https://doi.org/10.5281/zenodo.7409281. A readme file in the Data folder explains how to run the analyses to reproduce the results that we report in the paper. For any queries, please contact the lead author s.v.wass@uel.ac.uk.

The following datasets were generated:

| Author(s) | Year | Dataset title | Dataset URL | Database and Identifier |
| --- | --- | --- | --- | --- |
| Wass S | 2022 | Vocal communication is tied to interpersonal arousal coupling in caregiver-infant dyads | https://dx.doi.org/10.5061/dryad.612jm6473 | Dryad Digital Repository, 10.5061/dryad.612jm6473 |
| Wass SV | 2022 | Vocal communication is tied to interpersonal arousal coupling in caregiver-infant dyads | https://doi.org/10.5281/zenodo.7409281 | Zenodo, 10.5281/zenodo.7409281 |

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

# Appendix 1

## 1 Supplementary methods

### 1.1 Experimental participant details

This sample size was selected prior to the commencement of the study based on power calculations presented, and approved by peer review, in the funding application that supported this work (ESRC ES/N017560/1). Exclusion criteria included: complex medical conditions, skin allergies, heart conditions, parents below 18 years of age, and parents receiving care from a mental health organisation or professional. Full demographic details of the participants are given below.

**Appendix 1—table 1.** Demographic details for the sample (N=82).

| Infant age (days) – mean | | 351.9 |
|---|---|---|
| - *SE* | | 4.6 |
| Gender (% male) | | 39.3 |
| | | |
| Infant Ethnicity (%) | White British | 51.9 |
| | Other white | 11.4 |
| | Afro-Caribbean | 8.9 |
| | Asian, Indian & Pakistani | 10.1 |
| | Mixed - White/Afro-Carib | 2.5 |
| | Mixed - White/Asian | 7.6 |
| | Other mixed | 7.6 |
| | | |
| Household Income (%) | Under £16 k | 30.4 |
| | £16-£25 k | 29.1 |
| | £26-£35 k | 11.4 |
| | £36-£50 k | 12.7 |
| | £51-£80 k | 8.9 |
| | >£80 k | 7.6 |
| | | |
| Maternal education (%) | Postgraduate | 34.2 |
| | Undergraduate | 49.4 |
| | FE qualification | 2.5 |
| | A-level | 3.8 |
| | GCSE | 5.1 |
| | No formal qualifications | 2.5 |
| | Other | 1.3 |

### 1.2 Heart rate data

ECG was recorded at 250 Hz. To ensure good quality recordings, the ECG device was attached using standard Ag-Cl electrodes, placed in a modified lead II position. Due to technical problems with the ECG recording leads (N=9) and to problems with attaching the ECG recording electrodes securely (N=2), the ECG data were unavailable for 11 of the 93 participants originally tested.

To ensure the accuracy of these recording devices, they were cross-validated by recording heart rate and heart rate variability using both the new devices at home and established recording devices (a Biopac MP150 amp recording at 2000 Hz) in lab settings. High reliability was observed both for heart rate (rho = 0.57, p<0.001) and heart rate variability (rho = 0.70, p=0.01).

To analyse the Inter-Beat intervals, data were first parsed using a simple amplitude threshold (see e.g. *Aurobinda et al., 2016* for a similar approach), with R peaks identified as moments where the raw ECG signal exceeded the threshold value. Initially, the threshold value was set high; the same process was then repeated at incrementally decreasing thresholds.

At each threshold value, the R peaks identified were automatically subjected to the following checks. These threshold values were set following extensive piloting and visual inspection of our infant ECG data using the visualisation shown in Figure S2. (i) minimum temporal threshold: does the R peak occur at a time interval of greater than 300 ms since the previous R peak (corresponding to a heart rate of 200 BPM); (ii) maximum temporal threshold: does the R peak occur at a time interval of less than 850 ms since the previous R peak (corresponding to a heart rate of 70 BPM); (iii) maximum rate of change: when we calculate the R to R interval between this peak and the subsequent peak, and compare it with the R to R interval between this peak and the previous peak, is this difference less than 300 ms? In setting these threshold values, careful attention was paid to visual inspection to determine the maximum and minimum 'genuine' heart rates observed in our infant data; in setting the maximum rate of change criterion, careful attention was paid to identify the maximum rate of vagally mediated heart rate changes in infants.

Figure S2 shows a sample screenshot from the Matlab processing algorithm that was used. Two separate types of artefact are shown. The first, highlighted by the call-out figures at a and d, are instances where the ECG signal for a particular beat was lower than the threshold, and a genuine beat was missed. It can be seen that in both instances, the R peaks either side of this missing beat have been automatically identified, and excluded. These artifacts were identified based on the maximum temporal threshold criterion in example a and d, and additionally based on the maximum rate of change criterion in example d. The second, highlighted by the call-out figures at b and c, are instances where the ECG signal exceeded the amplitude threshold, and an incorrect R peak was identified. In both instances, the incorrect beat has been identified based on the minimum temporal threshold criterion, and the R peaks either side of this incorrect beat have been identified and excluded. Please note also that the sample below has been selected in order to demonstrate how the program identified the most common artefacts in the data. Overall, the occurrence of both types of artefact in our data is relatively rare, as is shown in Figure S3, below.

These three criteria were applied separately to data after it had been parsed at each threshold value. Following this, at each threshold value, the proportion of candidate R peaks that were rejected was compared with the proportion of candidate R peaks that passed all three criteria. The threshold value with the lowest proportion of rejected candidate R peaks was chosen as the threshold used for that participant.

In addition, and as a further check, a trained coder who was naïve to study hypotheses double coded a randomly selected subsample of 1000 beats for 20% of the participants, coding them as genuine or artefactual. Cohen's kappa was calculated to measure inter-rater reliability between the manual coding and the automatic coding, based on the best-fitting threshold level. This was found to be 0.97, which is high (*McHugh, 2012*).

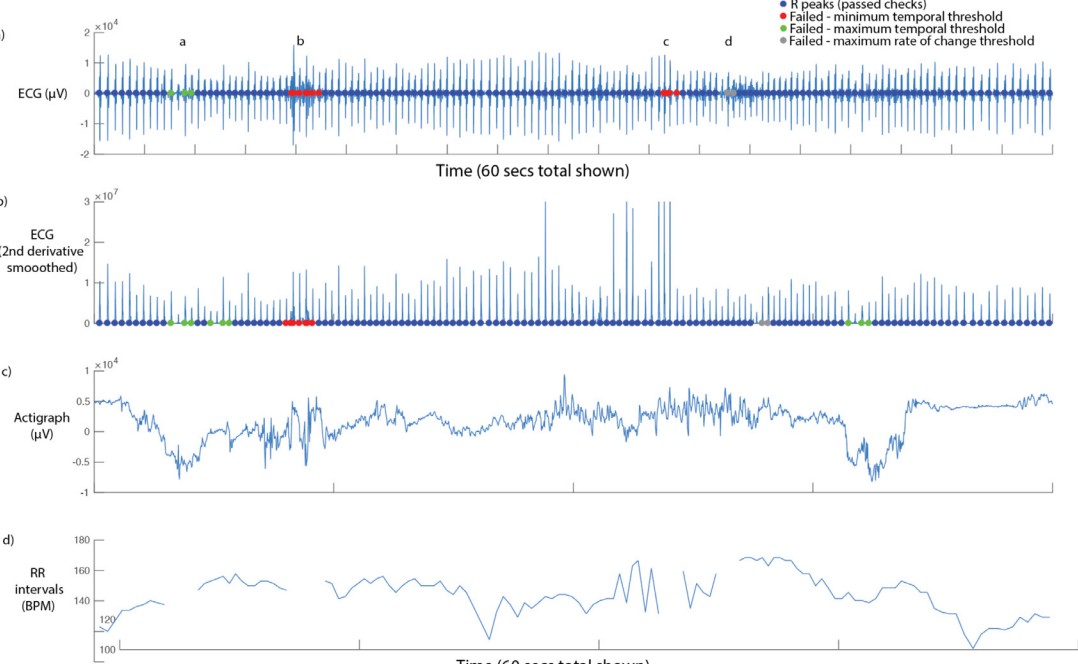

**Appendix 1—figure 1.** Sample screenshot from ECG parsing algorithm. 60 seconds' data is shown. From top to bottom: (i) raw ECG signal. Coloured dots show the results of the three checks described in the main text, below (see legend); (ii) smoothed second derivative of ECG signal. This measure was not used as our pilot analyses found it to be less effective than applying the processing to the raw signal; (iii) raw (unprocessed) actigraph data. This information was only used for visual inspection, and was not used in parsing; (iv) RR intervals (in BPM), with rejected data segments excluded.

Figure S2 below shows a histogram of the proportion of candidate R peaks rejected for each participant, based on the best-fitting threshold value. The median (st. err.) is 1.07 (0.36) % data rejected. This relatively low figure was achieved through very close attention during the piloting phase to the selection and placement of the ECG electrodes, to the design of the device, and the gain settings on the recording device.

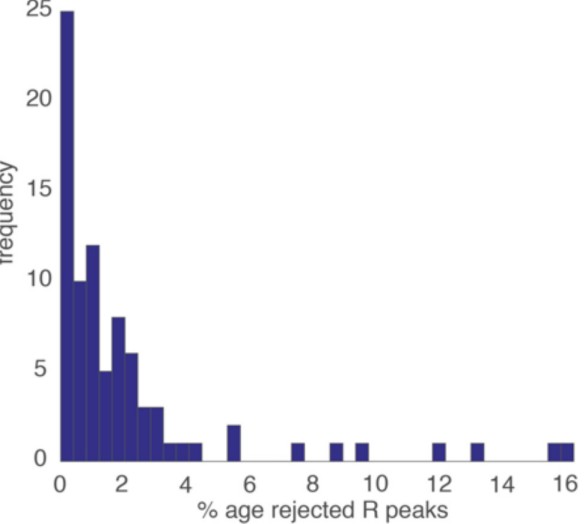

**Appendix 1—figure 2.** Histogram showing the proportion of rejected R peaks (as identified using the three criteria described above).

## 1.3 Heart-rate variability (HRV)

HRV was calculated using the PhysioNet Cardiovascular Signal Toolbox (*Vest et al., 2018*). In these scripts, which performed a completely separate analysis of the ECG data, a 60-s window with an increment of 60-s was implemented, and the default settings were used with the exception that the min/max inter-beat interval was set at 300/750ms for the infant data and 300/1300ms for the adult data. The Root Mean Square of Successive Differences (RMSSD) measure was taken to index Heart Rate Variability, but other frequency domain measures were additionally inspected and showed highly similar results, as expected (*Vest et al., 2018*).

## 1.4 Actigraphy

Actigraphy was recorded at 30 Hz. To parse the actigraphy data we first manually inspected the data, then corrected artifacts specific to the recording device used, then applied a Butterworth low-pass filter with a cut-off of 0.1 Hz to remove high-frequency noise, and then averaged from three dimensions into one. Actigraphy data were available for all participants tested.

## 1.5 Arousal composite

Previous research has shown significant patterns of tonic and phasic covariation between different autonomic measures collected from infants (*Wass et al., 2016*; *Wass et al., 2015*). Here, we include plots showing that the present dataset replicated and extended these results. The plots only show the sections of the data when participants were at home, comparing sections in which the infants were awake and asleep. Figure S2a shows cross-correlation plots examining the relationship between heart rate and movement. In both waking and sleeping sections the zero-lag correlation is 0.5. Figure S2c shows how these zero-lagged correlations vary on a per-participant basis. S2b shows an illustrative sample from a single participant. Sleeping sections show very low movement levels and lower heart rate. Of note, heart rate and movement do still inter-relate during the sleeping sections of the data (Figure S2c), albeit that the variability in heart rate and movement is lower. (Figure S2 d-f) show similar relationships between heart rate and heart rate variability, illustrating the strong and consistent negative relationships that were observed between these variables, as predicted.

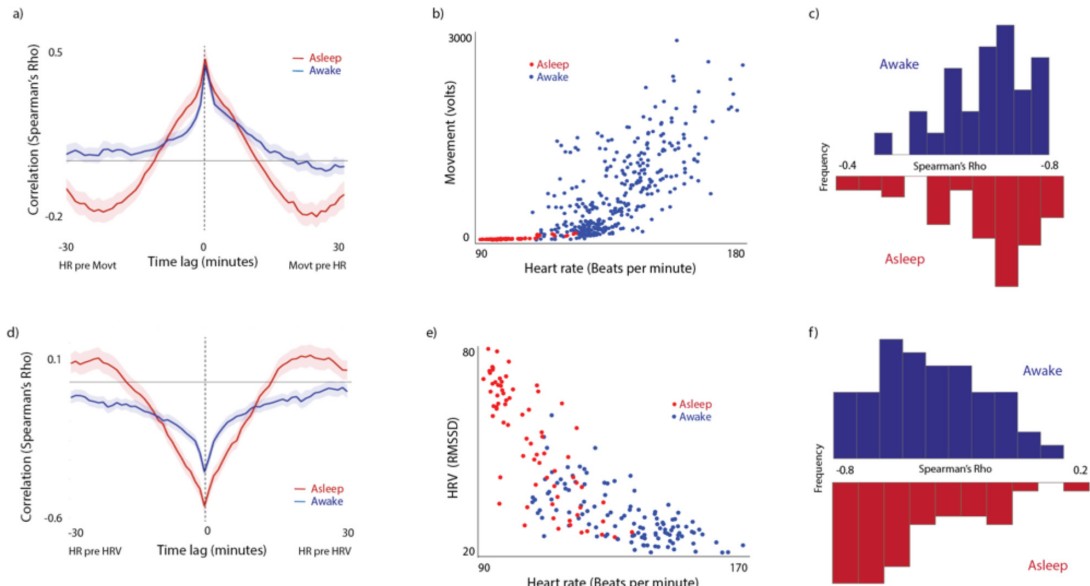

**Appendix 1—figure 3.** Illustrating the relationship between the individual physiological measures included in the composite measure. (**a**) Cross-correlation of the relationship between HR and Movement. (**b**) Scatterplot from a sample participant. Each datapoint represents an individual 60-s epoch of data. (**c**) Histograms showing the average zero-lagged correlation between 60-s epochs, calculated on a per-participant basis and then averaged. (**d**-**f**) Equivalent plots for Heart rate and Heart rate variability.

Extensive previous research has identified fractionation, and differentiation, within our autonomic response systems (*Jänig and Häbler, 2000*; *Kreibig, 2010*; *Lacey, 1967*; *Levenson, 2014*; *Quas et al., 2014*) – suggesting, for example that the sympathetic and parasympathetic subdivisions

operate, to an extent, in a non-additive manner (*Samuels and Szabadi, 2008*). Although indubitable, these findings should be seen as rendering incorrect our treatment here of autonomic arousal as a one-dimensional construct. Like many other arguments concerned with general versus specific factors, the question is rather one of the relative proportions of variance that can be accounted for by a single common factor in comparison with the variance accounted for by the sum of specific factors (*Graham and Jackson, 1970*) (see also *Calderon et al., 2016*).

As a result of these considerations, the three autonomic measures were collapsed into a single composite measure for all analyses. To do this, the actigraphy data was first subjected to a log transform (*Thomas and Burr, 2008*), to correct the raw results, which showed a strong positive skew (*Wass et al., 2016*) (see also SM section 1.6, below). Second, all three variables were converted to z-scores. Third, the HRV data were inversed because of the overall negative relationships noted between HRV and the other two measures (see Figure S4). Fourth, the three z-scores were averaged.

On the occasions where heart rate data were excluded due to artifact, data from actigraphy alone was used for the composite variable. Note that these occasions were relatively rare (accounting for a median ~=1% of all data - see Figure S3), and that the zero-lag cross-correlation between movement and heart rate across all available data was high (~=0.5 – see Figure S4).

## 1.6 Removal of autocorrelation from arousal data

Autonomic arousal data are known to show autocorrelation (*Wass et al., 2016*). In order to preclude the possibility that differences in the autocorrelation may have influenced results, the autocorrelation was removed from the data prior to performing all calculations, using the following procedure. First, best-fit bivariate polynomials were calculated for the two time series independently, in order to remove linear and quadratic trends, and the residuals obtained were subjected to the Dickey-Fuller test to check that they showed stationarity, which they did. The residuals were used in subsequent analyses. Next, in order to remove the autocorrelation component from each time series independently, univariate autoregressive models were fitted to each time series, and the residuals were calculated (see e.g. *Feldman et al., 1999*; *Feldman et al., 2011*; *Jaffe et al., 2001*; *Suveg et al., 2016* for similar approaches). The residual values (shown in *Figure 1*) were converted into z-scored values. These z-scored values were then used for all analyses. The only exception to this is the analyses specifically examining changes in autocorrelation relative to vocalisations, for which the raw uncorrected data were used.

## 1.7 Home/awake coding

### 1.7.1 Home/not home

Coding of when participants were at home was performed using the GPS monitors built into the recording devices. The position of the participant's home was calculated based on the postcode data that they supplied, and any GPS samples within a 50 m area of that location were treated as Home (corresponding to the accuracy of the GPS devices that we were using).

### 1.7.2 Sleeping/waking

To identify samples in which infants were sleeping, parents were asked to fill in a logbook identifying the times of infants' naps during the day. This information was manually verified by visually examining the actigraphy and ECG data collected, on a participant by participant basis. Actigraphy, in particular, shows marked differences between sleeping and waking samples (see *Figure 1* in main text), which allowed us to verify the parental reports with a high degree of accuracy. N=4 of the participants recorded did not sleep during the day that we were recording.

## 1.8 Video coding of physical positions while vocalising

In order to ascertain the physical position of our participants during the vocalisations we also performed an additional analysis based on the data recorded by the wearable Narrative Clip 2 Cameras worn by participants. These cameras were positioned in a breast pocket on the T-shirts worn by the infants (see *Figure 1* in main text). Like the microphones, they were programmed to record a 5-s video every minute. A trained coder watched each video one by one and categorised then into the following four categories: Free moving (camera is moving, and infant is self-generating movement); free stationary (camera is stationary and the baby is stationary but not strapped in); strapped sitting (child is strapped in to a feeding chair, pushchair, sling, car seat, shopping trolley etc); carried (camera is moving and the child id being freely carried in an adult's arms). Because

of its highly labour-intensive nature this coding was only performed for a subset of participants (N=24).

## 1.9 Permutation-based clustering analyses

To estimate the significance of the time-series relationships in the results, a permutation-based temporal clustering approach was used. This method examines temporally contiguous patterns of change in instances where the centre-point of the expected response window is unknown, or unimportant (*Maris and Oostenveld, 2007*). In each case, the test statistic (always specified in the text) was calculated independently for each time window. Series of significant effects across contiguous time windows were identified using an alpha level of.05. 1000 random datasets were then generated with the same dimensions as the original input data. To ensure that the same level of autocorrelation was present in the simulated data as in the original datasets, multivariate autoregressive models were fitted to each sample included in the original dataset using the Matlab function ARfit.m (*Neumaier and Schneider, 2001*), and the matching AR parameters were used to generate each of the random datasets using the Matlab function ARsim.m (*Neumaier and Schneider, 2001*). Then, the same sequence of analyses was repeated, and the longest series of significant effects across contiguous time windows was identified. The results obtained from the random datasets were used to generate a histogram, and the likelihood of observed results have been obtained by chance was calculated by comparing the observed values with the randomly generated values using a standard bootstrapping procedure. Thus, a p value of<.01 indicates that an equivalent pattern of temporally contiguous group differences was observed in 10 or fewer of the 1000 simulated datasets created.

## 2 Supplementary results

### 2.1 Supplementary analysis 1 – breakdown of vocalisations by type, affect and intensity

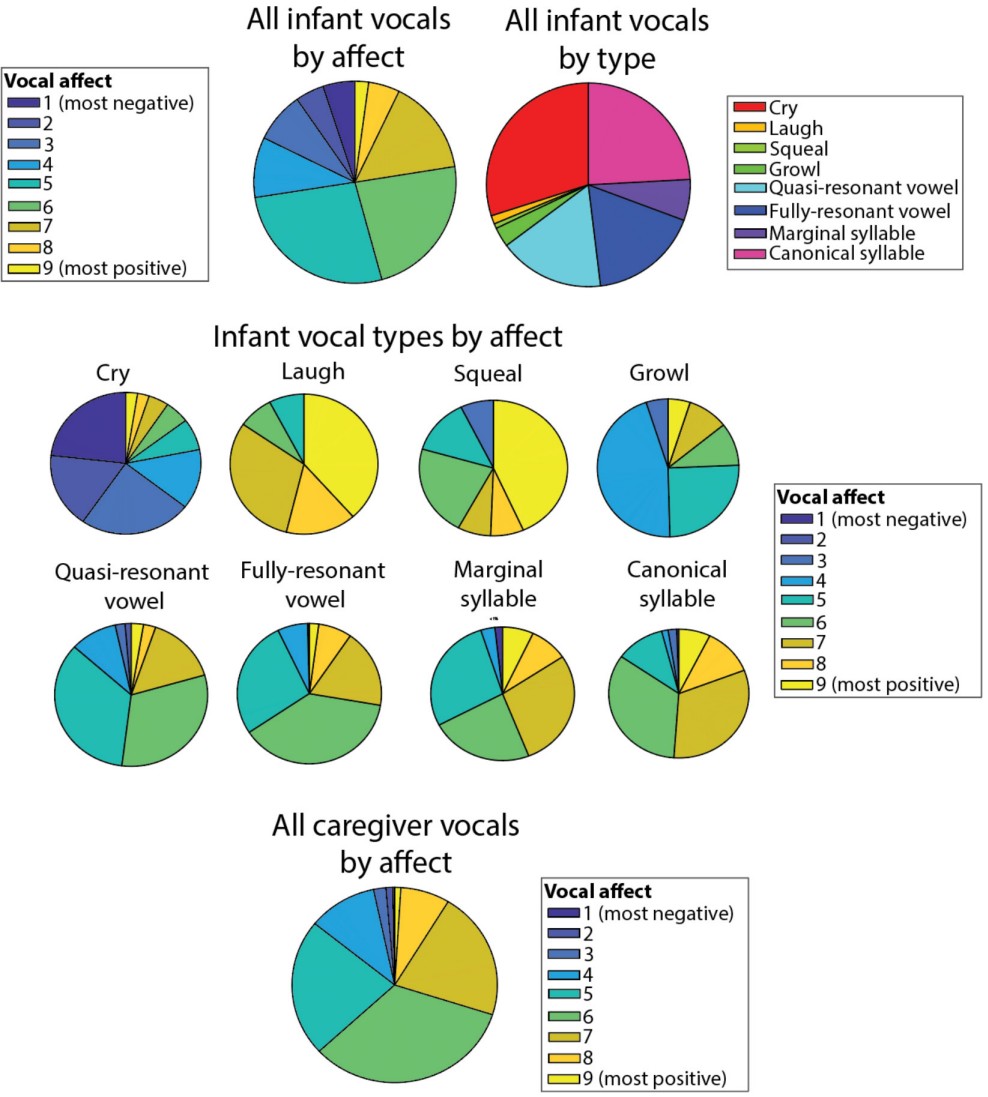

**Appendix 1—figure 4.** Pie charts showing infant vocalisation type by vocal affect.

### 2.2 Supplementary analysis 1 – video analyses of physical position while vocalising

One possibility is that the association we observed between vocalisations and arousal may be because vocalisations are more likely to occur in physical positions otherwise associated with increased arousal. To examine this, we performed an additional coding to examine infants' physical positions while vocalising. This was performed on the video footage recorded by the wearable cameras worn by participants, as described in section 1.8.

Overall our results suggested that the majority of vocalisations occurred while the dyad was either free moving or free stationary (*Appendix 1—figure 5a, b*)**Fig S5a,b**. For infant vocalisations, 49% of vocalisations occurred while free moving; 33% while free stationary; 7% while strapped sitting; 11% while carried. For adult vocalisations, 43% occurred while the infant was free moving; 33% while free stationary; 10% while strapped sitting; 12% while carried. 10/24 participants recorded no samples that were coded as 'strapped sitting', and 2/24 recorded no samples that were coded as 'carried'.

To analyse the data, we conducted Bonferroni-corrected pairwise comparisons between categories. Results of these comparisons are shown in *Appendix 1—figure 5*. Results significant at p<0.05 are indicated with a *. *Appendix 1—figure 5c, d* show the results of how infant and parent arousal differed by physical position. Only some categories show significant differences: for example, caregiver arousal was higher while the infant was being carried relative to when the infant was free moving, and when they were strapped sitting. No significant differences between physical positions were observed for either infant arousal auto-correlation (*Appendix 1—figure 5e*), caregiver arousal auto-correlation (*Appendix 1—figure 5f*), or infant-caregiver arousal cross-correlation (*Appendix 1—figure 5g*).

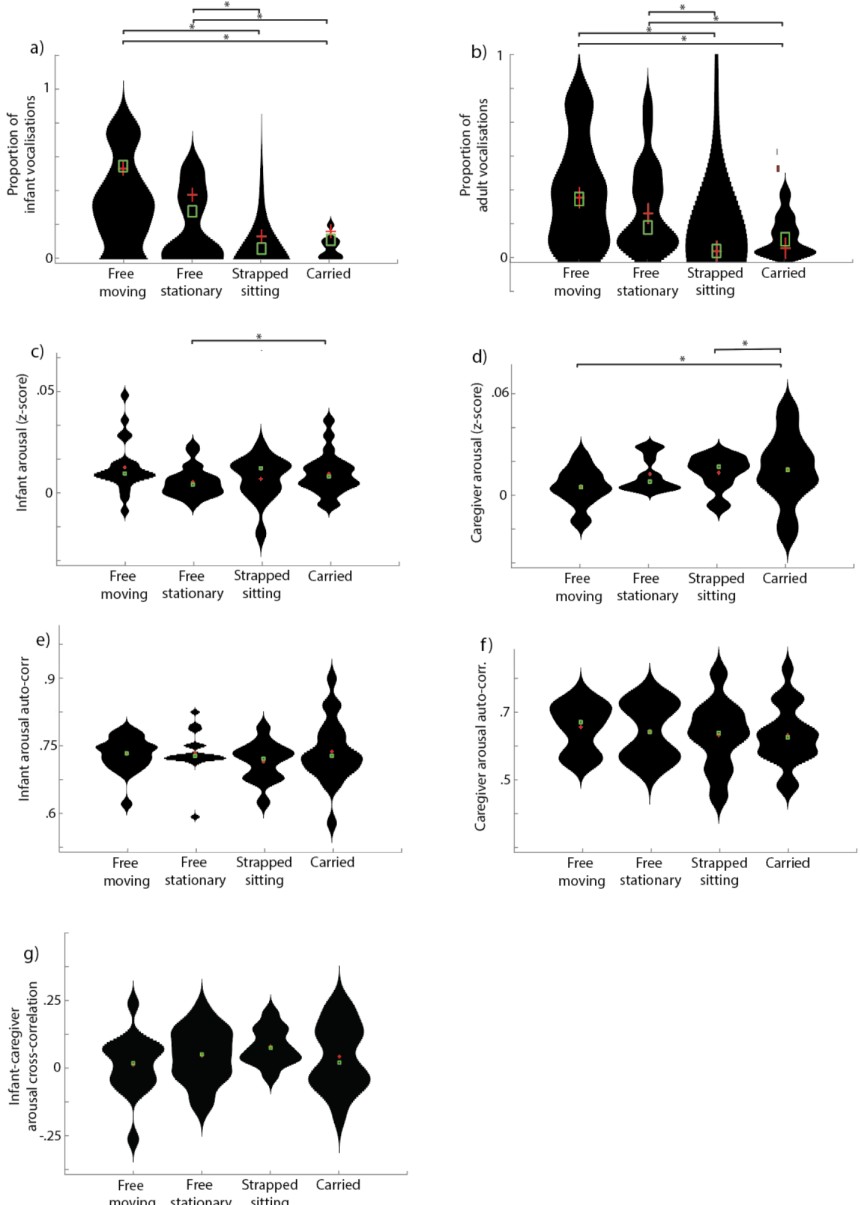

**Appendix 1—figure 5.** Violin plots showing results of physical position coding. (**a**) and (**b**) show proportion of infant vocalisations and proportion of adult vocalisations in each of the four physical positions coded. (**c**) and (**d**) show infant arousal in each of the four physical positions coded. (**e**) and (**f**) show arousal auto-correlation in each of the four physical positions coded. (**g**) shows infant-caregiver arousal cross-correlation in each of the four physical positions coded. For all analyses, * indicates significant pairwise post hoc between group comparison after correcting for multiple comparisons, p<0.05.

## 2.3 Supplementary analysis 2 - Comparison dataset with micro-level coding

Another possibility is that the arousal increases around vocalisations that we observed may be because the physical act of vocalising causes increases in arousal. To address this, we conducted a more fine-grained analysis on a different dataset in which we continuously recorded vocalisations and arousal in a cohort of 11-month-old infants (mean (*std*): 10.86 (*1.23*) years) during two 5-min infant-caregiver tabletop interaction. Timings of the onsets of vocalisations were recorded to an accuracy of 20 Hz (i.e. 50ms). Dual EEG were also recorded during these interactions, but these data are not presented here. These analyses were based on heart rate data alone, since the other measures included in the composite arousal measure were not available for this dataset. In total, 2270 adult and 623 infant vocalisations were recorded, which was a mean (*st.err.*) of 66.8 (*3.3*) adult and 18.3 (*2.6*) infant vocalisations per dyad. Mean (*st.err.*) vocalisation duration was 2.3 (*0.2*) and 1.3 (*0.1*) for infant and adult vocalisations respectively.

Figure S6a shows the change in arousal from 20 s before to 20 s after each vocalisation. It appears that, for infant vocalisations, infants' arousal levels start to increase before the onset of a vocalisation and to return to baseline approximately 20 s after. The same permutation-based cluster analyses as described in the main text were calculated to compare the observed values with a chance value of 0. No significant differences were identified. No increases in infant arousal were observed relative to caregiver vocalisations, and no increases in caregiver arousal were observed relative either to caregiver or to infant vocalisations. Overall these results suggest that, in a seated tabletop interaction, caregivers show no change in arousal relative to vocalisations either from themselves or their partner (the infant). Infants showed non-significant increases in arousal relative to their own vocalisations, which started to increase 5 s before a vocalisation and returned to baseline 20 s after. No changes in infant arousal were observed relative to caregiver vocalisations.

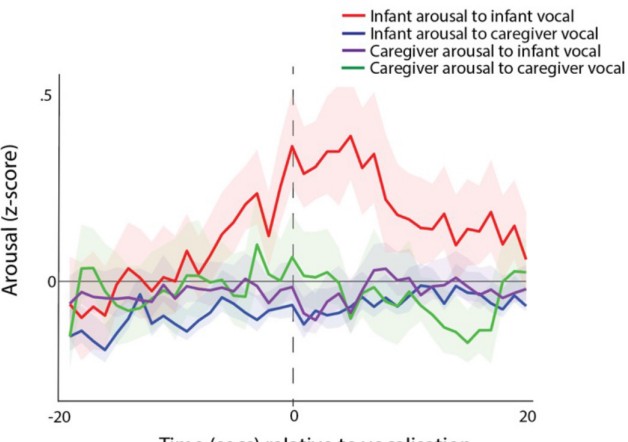

**Appendix 1—figure 6.** Arousal changes around vocalisations based on micro-level coding. (**a**) Same as *Figure 2b* in the main text, examining infant and caregiver arousal changes to infant and caregiver vocalisations. Shaded areas show standard errors.

## 2.4 Supplementary analysis to Figure 1f – vocalisation likelihood around arousal peaks

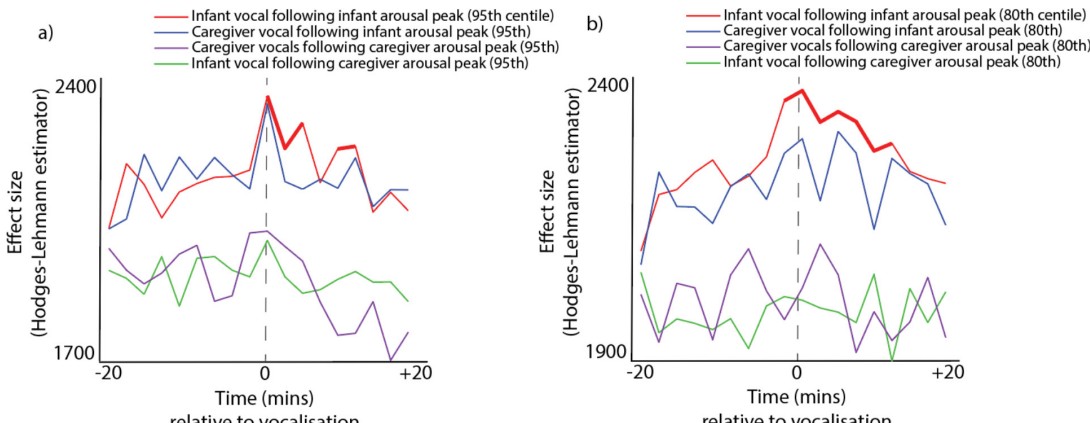

**Appendix 1—figure 7.** Identical to *Figure 1f* in the main text, except that different thresholds were used to define arousal peaks. (**a**) shows the analysis repeated relative to 95th centile arousal peaks; (**b**) shows the analysis repeated relative to 80th centile arousal peaks.

## 2.5 Supplementary analyses for part 2 – arousal by infant vocalisation affect and intensity

In addition to the analyses presented in Part 2 of the main text, which examine arousal by infant vocalisation type, we also conducted additional analyses subdividing infant vocalisations by affect and intensity.

*Appendix 1—figure 8 continued on next page*

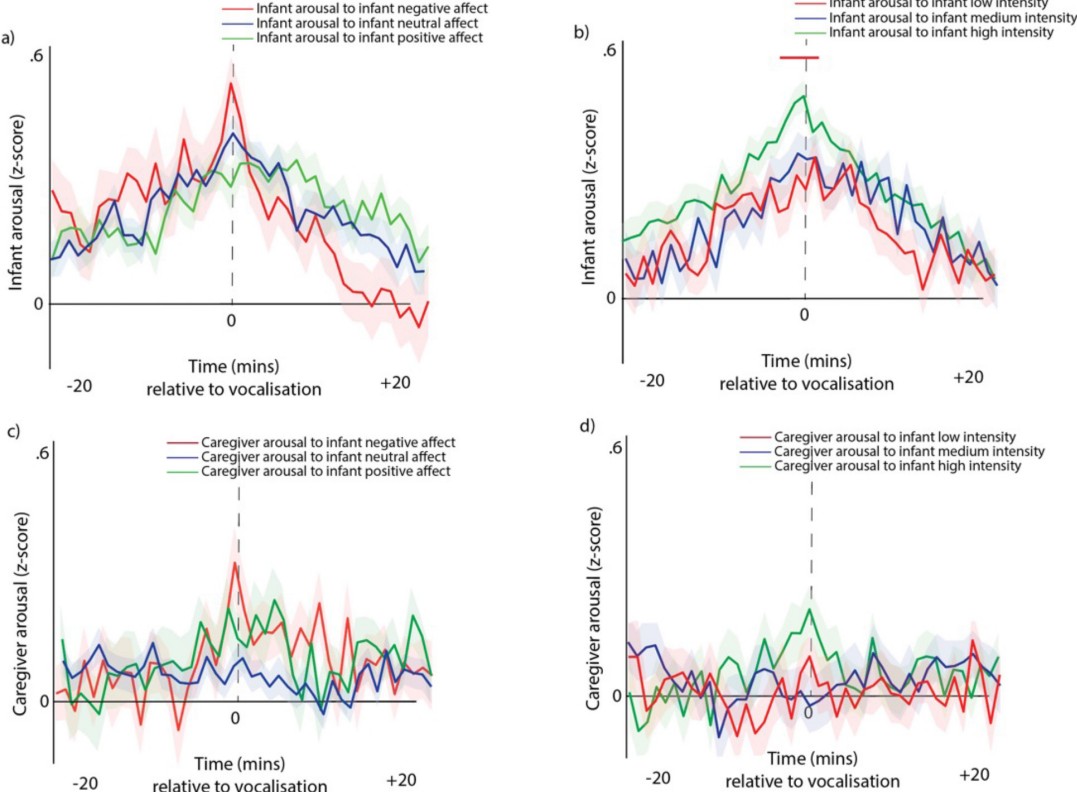

**Appendix 1—figure 8.** Arousal changes around vocalisations subdivided by infant vocalisation affect and intensity. (**a**) Infant arousal around infant vocalisations, subdivided by infant vocal valence. (**b**) Infant arousal around infant vocalisations, subdivided by infant vocal intensity. (**c**) Identical to a, but examining the change in caregiver arousal, subdivided infant vocal affect. (**d**) Identical to b, but examining the change in caregiver arousal, subdivided by infant vocalisation intensity. For all plots, shaded areas indicate standard error, and red highlights indicate areas of significant difference after correction for multiple comparisons.

## 2.6 Supplementary analyses for part 2 – arousal by caregiver vocalisation type and intensity

In this section we present further analyses subdividing caregiver vocalisations by vocalisation type and intensity. Equivalent analyses are not presented for caregiver vocal affect due to the rarity of negative affect caregiver vocalisations in our sample (see *Appendix 1—figure 4*).

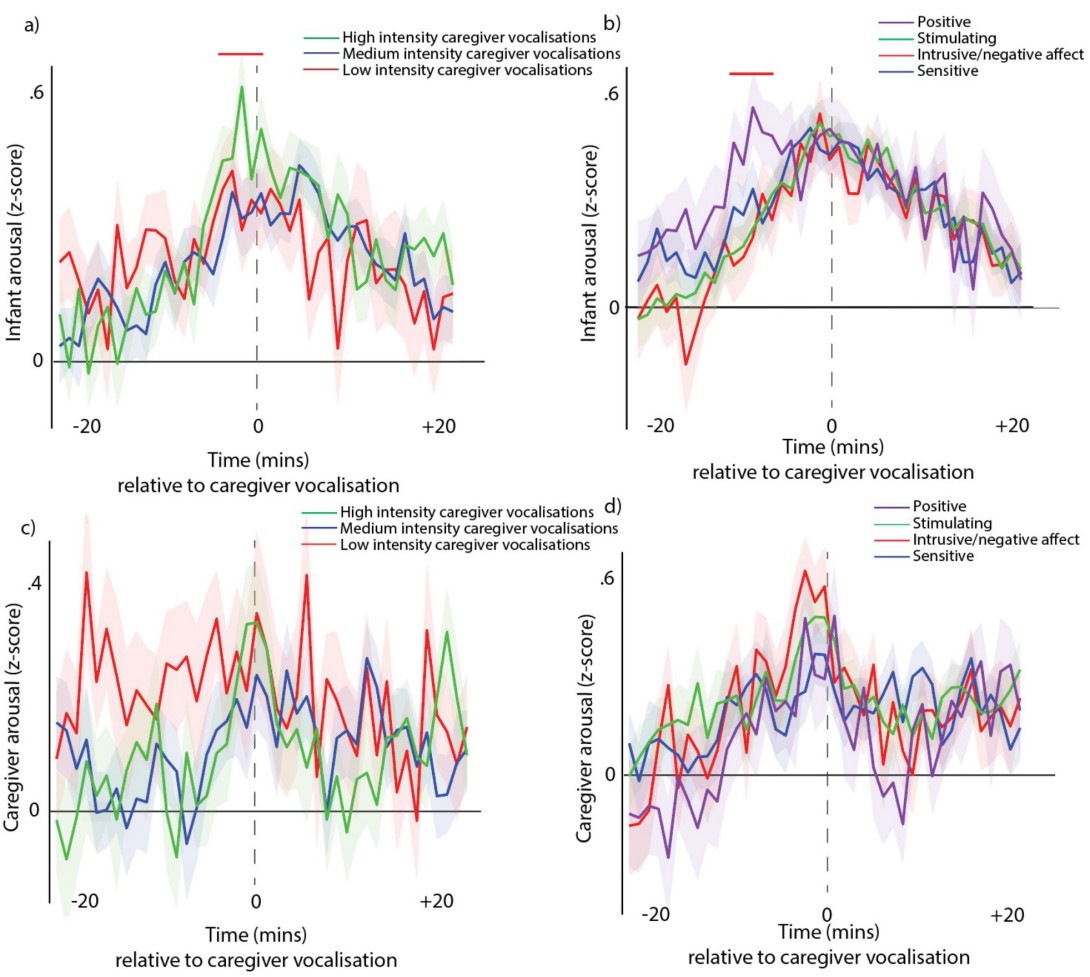

**Appendix 1—figure 9.** Arousal changes around vocalisations subdivided by adult vocalisation affect and intensity. (**a**) Infant arousal around caregiver vocalisations, subdivided caregiver vocalisation intensity. (**b**) Infant arousal around caregiver vocalisations, subdivided by caregiver vocalisation type. (**c**) Identical to a, but examining the change in caregiver arousal, subdivided caregiver vocalisation intensity. (**d**) Identical to b, but examining the change in caregiver arousal, subdivided by caregiver vocalisation type. For all plots, shaded areas indicate standard error, and red highlights indicate areas of significant difference after correction for multiple comparisons (Figs a and b only).

## 2.7 Supplementary analysis – simulation to examine the effects of sparse sampling on the data

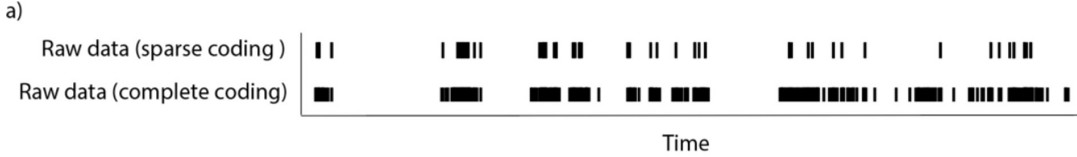

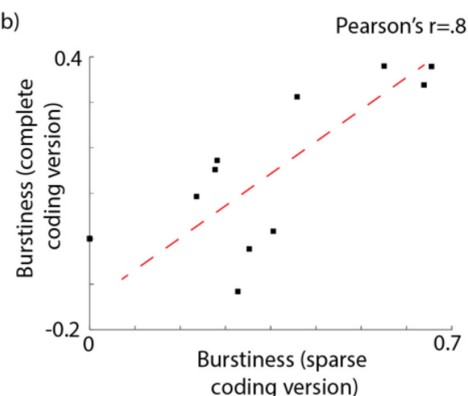

**Appendix 1—figure 10.** Examination of how sparse sampling affected the temporal distribution of our data. (a) Example raw data file comparing a 2-hr long segment of fully coded data (containing all vocalisations recorded, based on continuous recording) with a 'sparse coding' simulation (containing just the vocalisations recording during the first 5 s of every minute). (b) We obtained N=10 continuous hour-long recordings from 5- to 10-month-old infants and examined the temporal distribution of the data, comparing the continuous recording with the sparse coding simulation described in (b). To quantify the temporal distribution of the data we calculated the burstiness (following the equation used in *Abney et al., 2018*). Scatterplot shows the relationship between the burstiness as estimated from the complete coding version and from the sparse coding version. The Pearson's r between the two measures was r(9)=0.81, p<.001.

