## [Editor Report]

This study investigates how caregiver-infant communication is situated within (and drives) fluctuations in autonomic arousal using a cutting-edge methodology that combines day-long physiological measures and audio sampling. The authors report solid evidence on how caregiver and infant vocalisation in one-year-olds cluster around moments of heightened infant (and to a lesser extent caregiver) arousal. Overall, the article highlights the importance of examining physiological arousal in the study of caregiver-infant communication and speech development. The valuable descriptive findings and the potential of the novel methods used should be of interest to readers in the field of developmental science.

---

## [Decision Letter]

**Decision letter after peer review:**

Thank you for submitting your article "Interdependencies between vocal behaviour and interpersonal arousal coupling in caregiver-infant dyads" for consideration by *eLife*. Your article has been reviewed by two peer reviewers, and the evaluation has been overseen by a Reviewing Editor and Christian Rutz as the Senior Editor. The reviewers have opted to remain anonymous.

The reviewers have discussed their reviews with one another, and the Reviewing Editor has drafted this decision letter to help you prepare a revised submission.

Essential revisions:

As you will see in the reports appended below, both reviewers thought that this article represents an interesting contribution to the field, in terms of advancing our understanding of the development of communication abilities. At the same time, both reviewers agreed on several points that need to be addressed to strengthen the article and back up the conclusions that can be drawn. We highlight the most important points here:

1) The theoretical justification and articulation of the hypothesis and discussion need careful revision.

2) In addition, both reviewers agreed that some continuous data collection needs to be added to confirm the results from the sparse sampling. At least a couple of dyads are needed to confirm whether the sparse sampling is capturing a similar pattern to more continuous recording. A transcription of the speech is not needed -- time-stamping of the onsets and offsets of caregiver and child vocalizations would suffice. In fact, Reviewer #1 advised that LENA recorders (or some equivalent open-source software) can automatically time-stamp vocalizations (this is not perfect, of course, but it can provide a baseline).

3) Moreover, clarification of the code is necessary to ensure that results can be replicated.

*Reviewer #1 (Recommendations for the authors):*

Using an impressive dataset of at-home recordings of video, movement, and electrocardiogram data from caregivers and their infants, the authors examine how large-scale changes in infant arousal relate to spontaneous vocal production in caregivers and their infants. The paper is exhaustive with many analyses and the authors are commended for the Supplementary Materials, which address several alternative hypotheses to the results presented in the manuscript.

Nonetheless, the theoretical justification driving the analyses in this study could be better articulated. The authors appear to be seeking evidence for functional flexibility in one-year-old infants and are using arousal as a novel pathway to answer this question (as opposed to facial expressions). A clear and coherent explanation of what functional flexibility is would be helpful in the introduction. In addition, some set of hypotheses and alternative hypotheses to drive the data analysis in relation to functional flexibility would better ground the paper in theory.

While Figures 2, and 4 are clear and fairly straightforward to interpret, Figures 1 and 5 need particular attention. Consider Figure 1 (Lines 155-175). This attempt to demonstrate that vocalizations occur in temporal clusters is not intuitive and very difficult to interpret. For example, in lines 168-169, "infant vocalisations were significantly more likely to occur relative to an infant vocalisation from 16 minutes prior to 20 minutes after." Do infant vocalizations occur in clusters of 36 minutes in duration? Or is it that given any infant vocalization, there is a probability of a vocalization 16 minutes prior/20 minutes after (with silence in the intervening time). Although a simple measure of burstiness could be a direct way to measure the clustering of vocalizations, it is unclear how this temporal clustering relates to functional flexibility.

Due to technical constraints, the authors randomly sampled 5 seconds of audio every minute through their recordings. There is no continuous audio recording. While the authors have done the best they can, given the constraints, lines 127-128 are quite surprising: "the presence of undetected vocalisations can only have weakened any patterns of event-related change that we did observe." (and restated lines 540-541). If more data weaken the present findings, how are we to believe this paper is not simply reporting a spurious result due to sparse, random sampling of infant vocalizations?

The natural structure and timing of human vocalizations and vocal interactions occur across time periods longer than 5 seconds in duration. The authors provide little evidence that a random 5-second sampling every minute throughout the day can accurately reflect the clustering of infant vocal production, adult vocal production, and infant-adult vocal interaction. This is a serious concern.

While this is an interesting paper with several novel findings, it is severely limited by sparse audio collection. Further, the code provided is disorganized, not intuitively labeled, and insufficient for replicating the figures in the paper.

When presenting data in a figure, it is helpful to provide a mock figure which illustrates the possible outcomes of the question you are asking. Such an approach would greatly help a naïve reader in understanding the figures and would help structure the Results section into a theory-driven approach.

A clear demonstration that the random subsampling of audio accurately captures the diversity of vocal behavior of the infant and adult at home would be incredibly useful and greatly assuage concerns about the data. This would entail having a 7-hour (or whatever duration is comparable to the dataset presented) recording of (at least) 1 dyad (editorial note: at least 2 dyads recommended; see point [2] above), transcribing the entire day, then randomly sampling 5 seconds every minute. As an example, you can calculate the actual timing of clusters of vocalizations and compare it with the randomly sampled subset. Bootstrapping would go even further in helping to determine the possible range of error of the data collected. This can of course be repeated with the arousal data as well.

*Reviewer #2 (Recommendations for the authors):*

Wass et al. explored (1) the clustering of caregiver and infant vocalizations over time, (2) the temporal dynamics of arousal surrounding infants' and caregivers' vocal productions and auditory perceptions, and (3) the role of specific infant vocalization types in the coregulation of arousal in the dyad. This work found that caregiver and infant vocalization clustered around moments of heightened infant (and to a lesser extent caregiver) arousal. Infant cries and speech-like vocalizations were associated with different fingerprints in terms of average arousal, arousal stability, and arousal coupling between the caregiver and the infant. These data highlight the importance of examining physiological arousal in the study of caregiver-infant communication and speech development.

The manuscript has many strengths -- in addition to its cutting edge methodology, it provides detailed descriptions of the patterns observed in this study, includes thorough supplementary materials, and excels in capturing the dynamics of arousal and vocalizations in the naturalistic context of the child's everyday environment. However, the manuscript would benefit from a deeper discussion of the interpretation of these findings. Specifically:

(1) An important point to clarify is the relation between functional flexibility and communicative function. The sentence that starts on p. 4, line 87 (specifically, "…are largely tied to arousal fluctuations rather than communicative functions") suggests the interpretation that if infant vocalizations are functionally inflexible (that is, they only happen when the infant is in a specific affective state -- e.g., high arousal), they don't have a communicative function. It is possible that infant vocalizations operate on a switch that turns on when a certain arousal level is reached, but to make that claim we need to know whether the link between arousal and vocalizations remains unchanged when the caregiver isn't around and isn't responding for an extensive period of time (seems like that's not the case, based on p. 3, lines 60-63). However, it is plausible that signaling that the infant is in a high arousal state to caregivers is an important communicative function. Perhaps the flexibility shown by adults (vocalizing equally across different affective states) is a sign of the more sophisticated content of their vocalizations -- if infants don't have access to the same linguistic precision to express nuanced meaning in different vocalizations, they may use frequency of vocalization to communicate their internal state instead. If the infant's arousal is coupled with the caregiver's vocal production, it may be that arousal facilitates effective communication between the caregiver and the child and helps infants vocalize in moments when they are most likely to receive feedback. It is also important to discuss why the results for valence (in prior work) and arousal (current study) may be different -- these are very different constructs and the authors don't discuss what functional flexibility in terms of each of these dimensions would mean.

(2) It is also important to discuss the possible mechanisms that could result in the sustained elevated arousal surrounding infant and caregiver vocalizations. Currently, caregiver and infant vocalizations are described as standalone events (except in the very first section on temporal clustering), when in fact they likely are highly interdependent. These vocalizations occur in the context of social interaction, and as the authors pointed out, vocalizations occur in clusters. Therefore, the minutes following an infant vocalization are more likely to have more caregiver (and infant) vocalizations as well (and this may differ depending on the type of infant vocalization; e.g., cries vs. speech-like vocalizations). The presence of further vocalizations by the caregiver may sustain the infants' elevated autonomic arousal. Similarly, if an infant shows heightened arousal before a caregiver vocalization, that may either be due to the caregiver responding to the infant's heightened arousal or vocalization, or it may signal a prediction from the infant. It would be interesting to explore, even if just in a descriptive way, how conversation turn-taking may have contributed to the observed results.

(3) The paper uses three measures related to autonomic arousal. However, the text lacks discussion of the unique contributions of each of the three measures (average arousal, arousal stability, and arousal coupling) to answering the main research questions. The paper would benefit from a clear discussion of the authors' predictions in terms of these measures. When might we expect these measures to converge or diverge? How might increases or decreases in each measure impact infants' behavior or cognition? If the findings are the result of exploratory analyses, it's good that the authors aren't overstating the extent to which they had prior hypotheses. However, even so, it is important to frame the motivation for each variable and outline alternative hypotheses for each measure.

In addition to the above comments, I also wanted to add some smaller points that could strengthen the paper.

Abstract:

(1) I wonder if the abstract could open with a stronger sentence (p. 2, line 21) describing why it is important and interesting to understand the link between arousal and early vocal communication (beyond there being little work on the topic). This is such an exciting field, and the authors could do a better job of hooking the reader. (Note: this isn't a comment I feel strongly about, just a suggestion.)

(2) Functional flexibility should either be defined in the abstract (in half a sentence -- e.g., the authors do this in the introduction on p. 3, line 59) or the authors could perhaps describe the results without using this term, as it is a bit jargony.

(3) The last sentence of the abstract may be a bit misleading in terms of the conclusions of the paper, specifically "caregivers' differential responses to specific types of vocalizations is an important factor driving speech development; and that this selection mechanism which drives vocal development is anchored in our stress physiology." As it is currently written, this sentence suggests that this paper directly probes the link between caregiver responses to infant vocalizations and infants' speech development. While there is other work examining this question, the current investigation only examines the link between autonomic arousal (and related measures) and vocalizations but not speech development; we don't see changes in how children vocalize over time in response to caregiver feedback.

Introduction:

(4) It would be very helpful to add a paragraph on functional flexibility. This paragraph can include (1) a definition of what the authors mean by the term in the context of this specific investigation (there is a short definition on p. 3, line 59, but more elaboration would be useful), and (2) why the development of functional flexibility is important to understand. The authors touch on bits and pieces of this second point on p.4, lines 75-89, but currently, the paragraph misses some important questions. What would it mean if infants' vocalizations are (or aren't) functionally flexible? How would that change or inform theories of speech development?

(5) The authors mention that there are some limited studies on how autonomic arousal influences infant vocalization likelihood in human infants (p. 4, line 76, references 18-20). Could the authors elaborate more on these studies (it seems like they are mostly focused on respiration)? Highlighting how this study is situated in the human infant literature could emphasize its importance (not just novelty).

Results:

(6) Broadly, the Results section would benefit from clearer transition sentences at the start of each subsection and analysis explaining what question the analysis answers (e.g., the authors did a great job at this on p. 12, line 249; they also do this well at the end of some sections -- for example, p. 10, lines 211-212; it would be great to frontload these connections to the bigger hypotheses earlier in the section as well). It would be especially useful to remind the reader of the motivation behind each measure (see the earlier point in the Introduction section).

(7) To what extent are the three different measures (average arousal, arousal stability and coupling) correlated? (I know by definition they don't have to be correlated but in practice, it's possible they are.) Also, I know the authors did some multiple comparisons corrections, but I wonder if they could elaborate more on how they navigated having many different exploratory analyses and how they interpret diverging results in these different measures (again, see earlier points about explaining the different predictions for each measure).

(8) I had trouble understanding the paragraph on p. 14, lines 283-290. Don't some of the results (e.g., no changes in arousal before and after caregiver vocalizations -- lines 287-288) contradict the findings of the current study? Further, the point that the change in arousal is not due to the physical act of vocalizing is fair, but it may still be due to some physiological preparation associated with the motor planning of a vocalization.

(9) I was curious how fluctuations in caregiver and infant arousal throughout the day compare? Are caregivers overall "flatter" throughout the day? I wonder if that could explain some of the differences as well?

Discussion:

(10) As in previous sections, I think the paper would benefit from a more detailed discussion of (1) what each measure means for the broader questions of functional flexibility, caregiver-infant communication, and speech development, and why we might expect different results across measures, and (2) what the alternative explanations for the results may be.

Methods:

(11) Overall pretty detailed, it might be helpful to describe the ROC analysis in more detail in the Methods section and also briefly remind the reader in the Results section what the analysis shows conceptually. I know it is a very common analysis method, but I could imagine someone reading this and not knowing how to interpret it. I also couldn't find information on the thresholding process (Analysis 3, p. 12, lines 251-254).

---

## [Author Response]

Essential revisions:As you will see in the reports appended below, both reviewers thought that this article represents an interesting contribution to the field, in terms of advancing our understanding of the development of communication abilities. At the same time, both reviewers agreed on several points that need to be addressed to strengthen the article and back up the conclusions that can be drawn. We highlight the most important points here:1) The theoretical justification and articulation of the hypothesis and discussion need careful revision.

In response to this we have completely rewritten the introduction and made changes to the discussion, following the suggestions made by the reviewers. We describe these changes in our point-by-point responses to the reviewers below.

2) In addition, both reviewers agreed that some continuous data collection needs to be added to confirm the results from the sparse sampling. At least a couple of dyads are needed to confirm whether the sparse sampling is capturing a similar pattern to more continuous recording. A transcription of the speech is not needed -- time-stamping of the onsets and offsets of caregiver and child vocalizations would suffice. In fact, Reviewer #1 advised that LENA recorders (or some equivalent open-source software) can automatically time-stamp vocalizations (this is not perfect, of course, but it can provide a baseline).

In response to this we have collected the new data samples and conducted the additional analyses that the editor and the reviewers had suggested. We have taken these data from a new study that includes continuous home recordings, that is currently piloting in our lab. We have data from a cohort of 10 infants, of whom 8 were 5 months of age, and the remaining 2 were 10 months. Other than the fact that the microphone records continuously, the data analyses conducted are identical to those presented in our manuscript. The results of this new analysis strongly support our original findings, by independently replicating key elements – as we describe further in the point-by-point response to reviewer #1, below. We feel more confident in our findings as a result of adding these additional analyses, and we are extremely grateful for the suggestion.

3) Moreover, clarification of the code is necessary to ensure that results can be replicated.

In response to this comment we have completely re-written the instructions that we have provided along with the code, and added a readme file that explains in detail how to run analyses in order to replicate each of the data figures contained in the manuscript.

Reviewer #1 (Recommendations for the authors):Using an impressive dataset of at-home recordings of video, movement, and electrocardiogram data from caregivers and their infants, the authors examine how large-scale changes in infant arousal relate to spontaneous vocal production in caregivers and their infants. The paper is exhaustive with many analyses and the authors are commended for the Supplementary Materials, which address several alternative hypotheses to the results presented in the manuscript.Nonetheless, the theoretical justification driving the analyses in this study could be better articulated. The authors appear to be seeking evidence for functional flexibility in one-year-old infants and are using arousal as a novel pathway to answer this question (as opposed to facial expressions). A clear and coherent explanation of what functional flexibility is would be helpful in the introduction.

We thank the reviewer for this excellent suggestion. We have now considerably amended the introduction (pp.3-4) to better define functional flexibility, as follows:

“But what determines when infant vocalisations occur initially, and what their acoustic characteristics are? Are infants’ early vocal explorations constrained, and if so, how? One possibility is that vocal explorations follow a stochastic regime early on, and that infants’ explorations of their vocal tract possibilities produce a wide and unconstrained repertoire of outputs that is then narrowed down through the parental selection mechanism described above. Consistent with this idea, Oller and colleagues have proposed that a fundamental ability that supports the emergence of speech is functional flexibility ^1,2^. An individual has functional flexibility when at least some of their vocalizations can occur alongside variable underlying affective states, and are not tied to specific communicative functions (e.g., expressing distress). This ability is necessary for the establishment of a language system : it is because we can produce specific sounds to convey different meanings that arbitrary, symbolic systems can emerge ^2^. For instance, we can say “fine” to mean that we are happy to do what a partner suggested, or on the contrary, to signal irritation at the end of a conversation. In short, functional flexibility is a necessary condition for arbitrariness, a key feature of words that supports the emergence of conventional symbolic systems. By contrast, non-human primate vocalisations remain largely inflexible with respect to arousal even in adulthood ^3^ (although see ^4^).”

In addition, some set of hypotheses and alternative hypotheses to drive the data analysis in relation to functional flexibility would better ground the paper in theory.

In response to this point we have refined how we define our research questions and hypotheses on p.8, in order to make the link to functional flexibility more clear, as follows:

“First, are caregiver and infant vocalisations as inflexible with regard to arousal as those documented in non-human primates? I.e., do different types of vocalisation, such as cries and speech-like sounds, show different patterns of association with arousal across the infant-caregiver dyad? Our hypothesis was that even speech-like vocalizations remain relatively tied to fluctuations in arousal during infancy in contrast with adulthood. Second, do spontaneously occurring vocalisations during the day co-occur with specific patterns of arousal synchrony and co-regulation? Here, our hypothesis, in line with the parental selection mechanism, was that caregivers would track infants’ arousal fluctuations, and that as a consequence their vocalizations and arousal would be largely tied to their infants rather than their own.”

While Figures 2, and 4 are clear and fairly straightforward to interpret, Figures 1 and 5 need particular attention. Consider Figure 1 (Lines 155-175). This attempt to demonstrate that vocalizations occur in temporal clusters is not intuitive and very difficult to interpret. For example, in lines 168-169, "infant vocalisations were significantly more likely to occur relative to an infant vocalisation from 16 minutes prior to 20 minutes after." Do infant vocalizations occur in clusters of 36 minutes in duration? Or is it that given any infant vocalization, there is a probability of a vocalization 16 minutes prior/20 minutes after (with silence in the intervening time). Although a simple measure of burstiness could be a direct way to measure the clustering of vocalizations, it is unclear how this temporal clustering relates to functional flexibility.

In response to this comment we have substantially re-written the text in order to make it clearer how these results should be interpreted (p.8). We are grateful for the reviewer’s suggestions to use a burstiness measure instead – and are actually using this measure in some of the new analyses we describe below. However, it is not possible to use burstiness here as we need to be able to use the same measure to look at inter-relationships *across* the dyad (i.e. how the likelihood of an infant vocalisation changes during the time window around caregiver vocalisations) as *within* individuals (i.e. how the likelihood of an infant vocalisation occurring changes during the time windows around infant vocalisations). The burstiness measure allows us to calculate the latter type of relationship but not the former, and so would be unsuitable for this analysis.

Due to technical constraints, the authors randomly sampled 5 seconds of audio every minute through their recordings. There is no continuous audio recording. While the authors have done the best they can, given the constraints, lines 127-128 are quite surprising: "the presence of undetected vocalisations can only have weakened any patterns of event-related change that we did observe." (and restated lines 540-541). If more data weaken the present findings, how are we to believe this paper is not simply reporting a spurious result due to sparse, random sampling of infant vocalizations?The natural structure and timing of human vocalizations and vocal interactions occur across time periods longer than 5 seconds in duration. The authors provide little evidence that a random 5-second sampling every minute throughout the day can accurately reflect the clustering of infant vocal production, adult vocal production, and infant-adult vocal interaction. This is a serious concern.

In response to this point we have collected data from a cohort of 10 infants, part of a new ongoing longitudinal study. 8 of these infants were of 5 months of age, and the remaining 2 were 10 months. The devices used in this new study record continuous microphone data, along with the same autonomic measures as in the current study. All data were parsed in exactly the same way. For this analysis we only coded 60-minute segments of microphone recording from each dataset. In this way, we recorded a mean (std) of 72 (33) infant vocalisations per dyad, 63 (45) caregiver vocalisations; amongst the infant vocalisations, 22 (28) were speech-like vocalisations, and 4 (6) were cries.

Second, in order further to assess both the similarity of effects observed between the sparse coding and the continuous coding and the replicability of our results, we repeated what we consider to be the two most important analyses contained in the manuscript (Figure 2b and Figure 5c) using just the new N=10 hour-long datasets. These results are shown in Author response image 1. Figure (a) shows the original analysis (contained in the main text as Figure 2b), and figure (b) shows the identical analysis with the new pilot data. The analyses conducted were identical between the two sets of results. Figure (c) shows the original analysis (Figure 5c in the main text), and figure (d) shows the identical analysis with the new pilot data.

The results reveal a strong degree of similarity across the two sampling methods. This is striking particularly given the low N of the new analysis, the low numbers of vocalisations obtained in the infant speech-like vocalisation and infant cry categories, and the fact that these infants were younger. With the reviewers’ permission we would prefer not to include this analysis in the main published manuscript, as we feel that the low N, and the mixed age range of the data included, would not strengthen the manuscript.

**Author response image 1. sa2fig1:** 

Second, in order to examine how the sparse sampling affected the temporal distribution of the data, we conducted an analysis examining the timing of clusters, using the approach that reviewer 2 suggested. We examined the timings of the vocalisations observed in the new, pilot continuously recorded data, and we compared it with a resampling to match the sparse sampling data in the original study. We did this by retaining from the continuously recorded data only the vocalisations recorded during the first 5 seconds of every minute.In order to quantify how the temporal distribution of the data was affected by this change, we calculated the burstiness measure used by Abney and colleagues^5,6^, exactly as the reviewer suggests above. This offers a way to quantify the temporal distribution of the data with a single number. The formula for calculating burstiness is: B=μτ−στμτ+στ where μτ is the mean and στ is the standard deviation of the inter-stimulus intervals (i.e. the time intervals between vocalisations) ^7^. The same burstiness calculation was conducted on the raw data and the simulated sparse data.

In Figure S10, (a) shows an example of the raw data from a single participant, showing both the raw data and the simulated sparse data. (b) shows a scatterplot of the burstiness scores from the complete coding version, compared with the burstiness scores from the simulated sparse data. Similarity between the two sets of results was Pearson’s r(9)=.81, p<.001. These data have now been included in the supplementary figures, as Figure S10 in the SM.

While this is an interesting paper with several novel findings, it is severely limited by sparse audio collection. Further, the code provided is disorganized, not intuitively labeled, and insufficient for replicating the figures in the paper.

We believe that our new analyses fully address the reviewer’s concern about our initial sampling method. In response to this comment, we have also improved the labelling of the code and data presented, and added a readme file to the code explaining the exact procedures needed for replicating the figures in the paper.

When presenting data in a figure, it is helpful to provide a mock figure which illustrates the possible outcomes of the question you are asking. Such an approach would greatly help a naïve reader in understanding the figures and would help structure the Results section into a theory-driven approach.

We have spent quite some time attempting to produce the figure as the reviewer suggests, but were unable to produce a satisfactory one. Overall, we felt that including it would not increase the clarity of the text. Therefore, whilst we thank the reviewer for the suggestion, we hope that the reviewer will be satisfied that the textual changes that we have made in response to their other comments on this theme.

A clear demonstration that the random subsampling of audio accurately captures the diversity of vocal behavior of the infant and adult at home would be incredibly useful and greatly assuage concerns about the data. This would entail having a 7-hour (or whatever duration is comparable to the dataset presented) recording of (at least) 1 dyad (editorial note: at least 2 dyads recommended; see point [2] above), transcribing the entire day, then randomly sampling 5 seconds every minute. As an example, you can calculate the actual timing of clusters of vocalizations and compare it with the randomly sampled subset. Bootstrapping would go even further in helping to determine the possible range of error of the data collected. This can of course be repeated with the arousal data as well.

Please see our detailed response to this point raised above.

Reviewer #2 (Recommendations for the authors):Wass et al. explored (1) the clustering of caregiver and infant vocalizations over time, (2) the temporal dynamics of arousal surrounding infants' and caregivers' vocal productions and auditory perceptions, and (3) the role of specific infant vocalization types in the coregulation of arousal in the dyad. This work found that caregiver and infant vocalization clustered around moments of heightened infant (and to a lesser extent caregiver) arousal. Infant cries and speech-like vocalizations were associated with different fingerprints in terms of average arousal, arousal stability, and arousal coupling between the caregiver and the infant. These data highlight the importance of examining physiological arousal in the study of caregiver-infant communication and speech development.The manuscript has many strengths -- in addition to its cutting edge methodology, it provides detailed descriptions of the patterns observed in this study, includes thorough supplementary materials, and excels in capturing the dynamics of arousal and vocalizations in the naturalistic context of the child's everyday environment.

We thank the reviewer for these comments.

However, the manuscript would benefit from a deeper discussion of the interpretation of these findings. Specifically:(1) An important point to clarify is the relation between functional flexibility and communicative function. The sentence that starts on p. 4, line 87 (specifically, "…are largely tied to arousal fluctuations rather than communicative functions") suggests the interpretation that if infant vocalizations are functionally inflexible (that is, they only happen when the infant is in a specific affective state -- e.g., high arousal), they don't have a communicative function. It is possible that infant vocalizations operate on a switch that turns on when a certain arousal level is reached, but to make that claim we need to know whether the link between arousal and vocalizations remains unchanged when the caregiver isn't around and isn't responding for an extensive period of time (seems like that's not the case, based on p. 3, lines 60-63). However, it is plausible that signaling that the infant is in a high arousal state to caregivers is an important communicative function.

We thank the reviewer for raising this excellent point. We do cite already in the introduction research suggesting that infants also vocalise when they are alone – which does partially speak to this point. However, we agree that investigating whether the link between arousal and vocalisations remains unchanged when the caregiver isn’t around would be a fascinating question for future research, and we now mention this point in the directions for future research (p.25). The sentence mentioned by the reviewer also does not appear in the new version of the introduction, as we agree that this formulation was misleading.

Perhaps the flexibility shown by adults (vocalizing equally across different affective states) is a sign of the more sophisticated content of their vocalizations -- if infants don't have access to the same linguistic precision to express nuanced meaning in different vocalizations, they may use frequency of vocalization to communicate their internal state instead. If the infant's arousal is coupled with the caregiver's vocal production, it may be that arousal facilitates effective communication between the caregiver and the child and helps infants vocalize in moments when they are most likely to receive feedback.

This is another excellent point, that we have also been pleased to follow the reviewer’s suggestion by including in the discussion (p.25).

It is also important to discuss why the results for valence (in prior work) and arousal (current study) may be different -- these are very different constructs and the authors don't discuss what functional flexibility in terms of each of these dimensions would mean.

We thank the reviewer for this suggestion. This is clearly an important venue for future research, but at the moment we don’t want to speculate too much on this aspect because, at this stage, we are not sure that this discrepancy between our and Oller et al.’s previous work truly reflects differential flexibility for arousal and valence during infancy as opposed to stemming from methodological differences… We have now elaborated on this in the discussion:

“This discrepancy with previous findings might be due to a genuine difference in functional flexibility across arousal and valence, which is highly possible given the orthogonality of these two constructs (one can be happy, highly aroused and positive, and angry, highly aroused and negative). However, at this stage it remains equally possible that the discrepancy between our study and previous studies focusing on facial affects stem from methodological differences. Focusing on physiological measures might be more sensitive than relying on overt displays, and further research should try and rely on other measures (e.g., acoustic analyses of vocalizations) to try and disentangle”.

(2) It is also important to discuss the possible mechanisms that could result in the sustained elevated arousal surrounding infant and caregiver vocalizations. Currently, caregiver and infant vocalizations are described as standalone events (except in the very first section on temporal clustering), when in fact they likely are highly interdependent. These vocalizations occur in the context of social interaction, and as the authors pointed out, vocalizations occur in clusters. Therefore, the minutes following an infant vocalization are more likely to have more caregiver (and infant) vocalizations as well (and this may differ depending on the type of infant vocalization; e.g., cries vs. speech-like vocalizations). The presence of further vocalizations by the caregiver may sustain the infants' elevated autonomic arousal. Similarly, if an infant shows heightened arousal before a caregiver vocalization, that may either be due to the caregiver responding to the infant's heightened arousal or vocalization, or it may signal a prediction from the infant. It would be interesting to explore, even if just in a descriptive way, how conversation turn-taking may have contributed to the observed results.

Again this is an excellent point, and we have made textual changes to the discussion to incorporate it (pp.24-25). How turn-taking shapes these interdependencies is indeed an important venue for future research, and we hope to be able to investigate this issue further once data collection for the study we describe above in response to reviewer 1 (and that we now use to address issues related to sampling) is complete. This novel dataset will be more suited to this research question, as it is continuous.

“Nevertheless, future research based on continuous recordings would allow us to examine in more detail the role of turn-taking behaviours in communicative exchanges – examining, for example whether arousal facilitates effective communication between caregiver and child by making children more likely to respond to verbal initiations by the caregiver. Furthermore, it would also be interesting to examine whether the link between arousal and vocalisations remains unchanged even in the absence of the caregiver, or where the caregiver is unresponsive.”

(3) The paper uses three measures related to autonomic arousal. However, the text lacks discussion of the unique contributions of each of the three measures (average arousal, arousal stability, and arousal coupling) to answering the main research questions. The paper would benefit from a clear discussion of the authors' predictions in terms of these measures. When might we expect these measures to converge or diverge? How might increases or decreases in each measure impact infants' behavior or cognition? If the findings are the result of exploratory analyses, it's good that the authors aren't overstating the extent to which they had prior hypotheses. However, even so, it is important to frame the motivation for each variable and outline alternative hypotheses for each measure.

In response to this point we have rewritten the section on p.5 where we first introduce the concept of arousal coupling. We discuss how infant and caregiver arousal relates to vocal behaviour, and also how arousal coupling relates to vocal behaviour, but would prefer, given the word limits, to limit our discussion of the differences between arousal, arousal stability and arousal coupling within this manuscript, in order to help maintain focus, as the differences between these measures are discussed extensively elsewhere ^8,9^.

In addition to the above comments, I also wanted to add some smaller points that could strengthen the paper.Abstract:(1) I wonder if the abstract could open with a stronger sentence (p. 2, line 21) describing why it is important and interesting to understand the link between arousal and early vocal communication (beyond there being little work on the topic). This is such an exciting field, and the authors could do a better job of hooking the reader. (Note: this isn't a comment I feel strongly about, just a suggestion.)

We have changed this as requested.

(2) Functional flexibility should either be defined in the abstract (in half a sentence -- e.g., the authors do this in the introduction on p. 3, line 59) or the authors could perhaps describe the results without using this term, as it is a bit jargony.

We have changed this as requested, to avoid using this term in the abstract (e.g., “caregivers vocalizations show greater decoupling with their states of arousal”).

(3) The last sentence of the abstract may be a bit misleading in terms of the conclusions of the paper, specifically "caregivers' differential responses to specific types of vocalizations is an important factor driving speech development; and that this selection mechanism which drives vocal development is anchored in our stress physiology." As it is currently written, this sentence suggests that this paper directly probes the link between caregiver responses to infant vocalizations and infants' speech development. While there is other work examining this question, the current investigation only examines the link between autonomic arousal (and related measures) and vocalizations but not speech development; we don't see changes in how children vocalize over time in response to caregiver feedback.

We have rewritten the last sentence of the abstract following this comment from the reviewer.

Introduction:(4) It would be very helpful to add a paragraph on functional flexibility. This paragraph can include (1) a definition of what the authors mean by the term in the context of this specific investigation (there is a short definition on p. 3, line 59, but more elaboration would be useful), and (2) why the development of functional flexibility is important to understand. The authors touch on bits and pieces of this second point on p.4, lines 75-89, but currently, the paragraph misses some important questions. What would it mean if infants' vocalizations are (or aren't) functionally flexible? How would that change or inform theories of speech development?

We have now completely rewritten the introduction to better define functional flexibility following previous authors, and to specify why this is important for speech development (see also above in response to reviewer #1).

(5) The authors mention that there are some limited studies on how autonomic arousal influences infant vocalization likelihood in human infants (p. 4, line 76, references 18-20). Could the authors elaborate more on these studies (it seems like they are mostly focused on respiration)? Highlighting how this study is situated in the human infant literature could emphasize its importance (not just novelty).

We have added a section on this, as requested (p.4).

Results:(6) Broadly, the Results section would benefit from clearer transition sentences at the start of each subsection and analysis explaining what question the analysis answers (e.g., the authors did a great job at this on p. 12, line 249; they also do this well at the end of some sections -- for example, p. 10, lines 211-212; it would be great to frontload these connections to the bigger hypotheses earlier in the section as well). It would be especially useful to remind the reader of the motivation behind each measure (see the earlier point in the Introduction section).

We have made these changes as the reviewer requests (see p.7 ff).

(7) To what extent are the three different measures (average arousal, arousal stability and coupling) correlated? (I know by definition they don't have to be correlated but in practice, it's possible they are.) Also, I know the authors did some multiple comparisons corrections, but I wonder if they could elaborate more on how they navigated having many different exploratory analyses and how they interpret diverging results in these different measures (again, see earlier points about explaining the different predictions for each measure).

See our response to this point above.

(8) I had trouble understanding the paragraph on p. 14, lines 283-290. Don't some of the results (e.g., no changes in arousal before and after caregiver vocalizations -- lines 287-288) contradict the findings of the current study? Further, the point that the change in arousal is not due to the physical act of vocalizing is fair, but it may still be due to some physiological preparation associated with the motor planning of a vocalization.

We have added text here (p.15) to explain these conclusions better.

(9) I was curious how fluctuations in caregiver and infant arousal throughout the day compare? Are caregivers overall "flatter" throughout the day? I wonder if that could explain some of the differences as well?

Quantifying this difference is complex given the multiple time-scales of change that we are examining in this paper. We would argue that the consistency of our findings across multiple analyses (including e.g. the findings of how vocalisation likelihood change around naturally occurring arousal peaks) – Figure 2e – indicate that this reason alone is unlikely to be an explanation for our results.

Discussion:(10) As in previous sections, I think the paper would benefit from a more detailed discussion of (1) what each measure means for the broader questions of functional flexibility, caregiver-infant communication, and speech development, and why we might expect different results across measures, and (2) what the alternative explanations for the results may be.

We thank the reviewer for raising these excellent points, and have addressed them in the revised discussion, as documented in our line-by-line responses above.

Methods:(11) Overall pretty detailed, it might be helpful to describe the ROC analysis in more detail in the Methods section and also briefly remind the reader in the Results section what the analysis shows conceptually. I know it is a very common analysis method, but I could imagine someone reading this and not knowing how to interpret it. I also couldn't find information on the thresholding process (Analysis 3, p. 12, lines 251-254).

We have added this information as requested to the method (p. 34), results (p.12) and the discussion (p.24).